# Bridges over the River Turia: Genesis of the Urban History of Valencia

**María-Montiel Durá-Aras \***, **Eric Gielen**, **José-Sergio Palencia-Jiménez** and **Josep Lluís Miralles-García**

School of Civil Engineer, Department of Urbanism, Universitat Politècnica de València (UPV),
46022 Valencia, Spain; egielen@urb.upv.es (E.G.); jpalenci@urb.upv.es (J.-S.P.-J.); jlmirall@urb.upv.es (J.L.M.-G.)
\* Correspondence: maduar@cam.upv.es; Tel.: +34-963877150

**Abstract:** The foundation of the city of Valencia was created by the Romans on an island formed by the River Turia, strategically located between *Carthago Nova* (Cartagena) and *Tarraco* (Tarragona), and is directly connected to the sea. This raises the question of how the elements of access to the city came about and how the river and its bridges might have affected its evolution. This article delves into the study of the origins of the city, with a time frame that extends into the 11th century, the time at which an event took place that confirms one of the major changes in the city's urban morphology: when it stopped being an island. The intrinsic relationship that exists between bridges and main communication routes as fundamental elements to the access of an island is the driving force behind this article, which is based on research, until now undone, on the existence and construction of the first bridges in the city of Valencia and their influence on the city's subsequent development. This paper will start by studying the founding and location of this city and will then analyze the communication routes existing at the time. It will also study the communication routes that were created later and how all of them were forced to cross a fluvial accident, the River Turia. For this purpose, the number of bridges built until the city ceased to be an island have been identified, and analyses of their typology, location and who was responsible for them has been carried out to study how they may have affected the normal flow and evolution of the riverbed and their possible influence on the city's development.

**Keywords:** barrier; river; flood; city; expansion; origin; geomorphology; bridge; connection





## 1. Introduction

The building of bridges in the ancient city of Valencia cannot be understood without analyzing its origins. This article was written as an attempt to discern the existence of the first bridges in the city of Valencia and how they influenced not only the development of the city, but also the creation of new communication routes. The bridges, which apparently endured the elements, ended up being ephemeral due to the vicissitudes of history and leading any urban planning process that involves a riverbed.

In the case of Valencia, founded by the Romans in 138 BC, bridges were built due to the importance of the Iberian Peninsula at that time in regards to war and conquest, particularly the Carthaginian lands. This is for the reason that once they were conquered, a strategic plan to dominate and control the east of the peninsula could be put into practice, therefore allowing them to encompass the whole of the Mediterranean. As in most cases, the roads existing then (and in the specific case of the Via *Herculea*), which served for the development, expansion, communication and business of Iberian society, provided the Roman army with a strong engineering spirit, with ways of entering and conquering new territories. As Ferran Arasa might say, the Roman Empire would never have been what it was without the wagons and the enormous constructive capacity of the Romans, and without Rome, roads would never have reached the civilizing level they had in history [1].

The Romans were true experts at 'crossings' or bridges, which are elements to link and safeguard against obstacles as part of the process of building new and reinforced

communication routes. In the case of the city of Valencia, where, prior to its foundation, the Via *Herculea* ran along its periphery as a communication route, there must have been crossings that made it possible to traverse the ancient River *Tyris* (Turia). The bridges used in the past would have been destroyed in some cases, by the force of the river, as will be discussed below. Subsequently, and taking into account the Romans studied location of the city, the bridges were either strengthened or moved, improving and creating new communication routes that would be of interest for the rest of the territory.

From the point of view of urban planning, this document analyzes the first crossings built by the Romans to create the new city founded under the name of *Valentia Edetorum*. It also analyzes how these influenced the creation of new engineering works, as well as the method of establishing and organizing cities amongst the Romans. We will continue with the evolution of the city after discussing different colonizing cultures, such as the Visigoths in 413 AD and the later Muslim invasion in 718 AD. We will take an analytical pause to look at the building of bridges, their necessity, durability and transformation throughout the first centuries of the country's history. All of this is due to the main transformation of the watercourse, which is a result of the loss of the right offshoot in its last stretch and potentially human. This has caused morphological and urban changes in the surrounding area.

Currently, there is hardly any research related to the city's first bridges, possibly because it is difficult to have written or graphically documented them until the 13th century. It is known that the Romans accessed the area through the Via *Herculea*, but its exact route around the island and the location of the first pass that crossed the River Turia are unknown. Once the city was founded and its location was known, there is no record of the number of steps that the Romans created, the order of their construction or their function and connection with the main roads. As far as roads are concerned, we can stick to the latest discoveries made in the rehabilitation works of the Palau de Benicarló in 2009 directed by Concha Camps [2], in which part of the Via *Augusta* was exposed on the outer west side of the Salvador Street. The gaps are also significant, with there being a few finds on the northern bank of the river that rule out its passage in the first section of Sagunto Street of the *Vía Augusta*, as indicated by the discoveries of Miguel Mañez [3] in 2002. Also, if we looked at the works on Ruaya Street in 2008, they would be linked to the layout of the *Vía Heráclea* as Carmen Aranegui [4] points out. In the set of evaluations there is little information about the passes or bridges, leaders of any route worthy of being exalted "bridgeheads" as Teixidor says [5]. Of the scarce data, highlight the testimony of Ihn Al-Qardabus in Loci de Abbadidis and some findings such as those exposed by Albert Ribera [6] in the extension of Corretgería Street with Reina Square.

The temporal scope of this study is limited to the period between the 1st century BC and the 11th century AD, covering the origins of the city until the disappearance of one of the arms of the river, at which time Valencia stopped being an island. Although the construction, transfer and updating of bridges has influenced the morphological changes that the city of Valencia has experienced until today, the influence that the first bridges had is especially relevant not only in the decision to locate the first city, but also in the original city plot and the consequent impact on the environment.

During this specific period, the existence of the first bridges in Valencia is analyzed to see how they could have influenced the initial urban development of the city. The bridges and their construction period, as well as their location, are identified. It is logical to think that a single bridge can feed the communication of a city located on an island and even more so in its beginnings, when the population density is low and its priority is the establishment of a city. However, due to the motivation for its creation and the influence of the environment, the creation of bridges becomes much more than simply safeguarding a riverbed. The bridge will influence its surroundings and its effects will last over time, conditioning the urban model.

In this context, a series of working hypotheses are proposed. The first of them is that there is **an existence of a relationship between the location of the bridges and the**

**population of the city**. To do this, a historical study will be carried out that will analyze both the strategic interests of the conquest and the importance and relationship of the main communication routes: the Via *Herculea* and the Via *Augusta*. Another fact to take into account is the geomorphological conditions of the terrain, which are the characteristics of the island. These include the riverbed that borders it and, very important at that time, its proximity to the sea. Then, in a second phase, we reflect on the origin of the bridges and their location. The existence of three bridges or steps is assumed. The **first step** used by the Romans at the time of the conquest that would form part of the Via *Herculea,* and would facilitate access to the island and area where the new city could be established. This first step would generate two sub-hypotheses, the possible layout of the Via *Herculea* and the possible location of the first bridge belonging to said route and not built by the Romans but used by them in their colonizing phase. The **second step** would also generate two sub-hypotheses, its authorship and its location. This step would safeguard one of the arms of the river channel and form part of the new route of the Via *Augusta*. Finally, the **third step**, called in some writings as *Al-Qantara* Bridge, puts forth two other sub-hypotheses about its possible location, with the first being its placement upstream of the ancient Roman port and the second, downstream of said port.

Finally, as Buhlers [7] said, bridges have the power to necessarily direct certain communication routes and manage to influence the design and creation of urban spaces. Also, because the bridge responds to a necessary function for human activities, it has been present since man has organized its itineraries. A final hypothesis is worked on related to the influence of each of them during the period in which they were in force in the urban model.

**2. Materials and Methods**

The methodology used was, due to the temporal nature of the study, based fundamentally on compilations of historical data referring to periods of time and archaeological evidence that confirms hypotheses or suppositions, both personal and cited from previously published books and articles. Both construction procedures and hydraulic theories have been added to this process, which have allowed proving the hypotheses of the possible location and existence of the first bridges.

For the analytical study of bridges, we have taken into account the *vectors of bridge design* [8]: the '*topos*', which is the study of the ideal location, the '*typos*', as a construction methodology taking into account the knowledge of the time period, and the '*form*', which is the expression of the definitive shape in terms of the design defined by the person who built it. In this way, the city's first bridges will be located, and their useful life will be analyzed according to geomorphological, social and urban evolutionary processes. It is in the process of siting on these bridges, where the need arises to intertwine the historical, the archaeological and the scientific, as if the pieces of a puzzle were trying to decipher the image of an urban plot headed by them.

Below is a list of the main techniques and data used:

| *MIXED METHODOLOGY STUDY OF BRIDGES* | **QUANTITATIVE** | 1- *Via* Herculea/*Via* Augusta *study*<br>2- *Geomorphological studies*<br>3- *Study and analysis of floods*<br>4- *Demographic study*<br>5- *Archaeological evidence*<br>6- *Roman bridge construction processes*<br>7- *Administrative documentation* |
|---|---|---|
| | **QUALITATIVE** | 8- *Historical documentation*<br>9- *Travellers' letters*<br>10- *Graphical documentation*<br>11- *Related documentation (doctoral theses, articles, publications, etc. . .)* |

## 3. Results and Discussion

### 3.1. Origins of the City of Valencia

The city of Valencia was founded, as is known, by the Romans in 138 BC. In their effort to conquer the Iberian Peninsula, they fought the Carthaginians in the 3rd century BC, using the existing roads created by the Iberians as a means of access. One of these roads, and possibly the most important, was the Via *Herculea*, which ran along practically the entire eastern part of the peninsula, from the Pyrenees to *Gades* (Cádiz). Archaeological discoveries attributed to the remains of roads prior to the Via *Augusta* confirm the existence of this Herculean Road in pre-Roman times, known by the Greek Author Timaeus of Tauromenium as the *Way of Hercules*, Antonio Sáez [9] echoing it.

It was this road that was used by the Romans to enter and conquer the Iberian Peninsula, one of their strategies being the extension of the Via *Herculea to Carthagonova*. This made it possible to control the eastern third of the peninsula and to dominate maritime traffic in the Mediterranean, placing three more ports under Italic authority (*Tarraco*, *Cartaghonova* and the recent *Valentia*).

There were several factors that led to the construction of various Roman engineering works. Of particular note are the trade and promotion of the state post office introduced in the 4th century BC, the foundation of new cities (as in the case of *Valentia*) and the division of large areas of land for distribution among the retired legionaries. In addition, these roads served as a tool to promote governance, which was responsible for leaving its 'mark' on these roads (milestones, bridges, etc.). So much so, in reference to Isaac Moreno [8], the construction of the Via *Augusta* was part of a territorial reorganization project set in motion by Emperor Augustus (hence its name) in the Hispanic provinces after winning the war against the Cantabrians and Asturians [10]. The Via *Augusta* is considered to be one of the most ambitious and emblematic projects carried out by the emperor in these lands. This road, which is approximately 1500 km long [11], allowed for the foundation of new cities and access to existing ones such as *Caesaraugusta* (Zaragoza), *Emerita Augusta* (Mérida) and *Barcino* (Barcelona), initiating an extensive and valuable network of road junctions.

The location of the city of *Valentia* is based on strategic criteria for the establishment of the Roman culture of the time. Its location on marshy land that was easy for the Romans to farm and in the middle of an elevated island surrounded by the two branches of the River Tirys towards the sea, and it being equidistant (approx. 250 km) from two major cities conquered by the Romans, *Tarraco* (Tarragona) and *Carthagonova* (Cartagena) [6], reinforce the theory of strategy and Roman cultural dominance of the time (Figures 1 and 2).

Originally, *Valentia* was a small village known as the nerve centre of the Levant, and soon became a colony despite being surrounded by other ancient cities: *Arse* (Sagunto), *Edeta* (Llíria), *Saetabis* (Játiva) and *Dianium* (Denia) (Figure 3).

Regarding the foundation of *Valentia*, the Via *Herculea* was, at that time, the only road that connected with the rest of the cities. The advantages of the city's surroundings were more than evident to the Romans, such as the proximity to the coast, the existence of a perennial river course with the possibility of navigation as well as a terraced base in the northern surroundings of a lagoon area [6]. As for its proximity to the sea, Ribera i Lacomba describes it by referring to Pliny when he narrates that 'the colony Valentia was 3000 paces from the sea' (equivalent to 4.5 km) [13]. This is important because today the area occupied by the Roman city is 5.5 km from the sea, 1 km more than 2000 years ago, indicating a slow but steady advance of the coastline produced by the Turia's alluvial sedimentation (Figure 4).

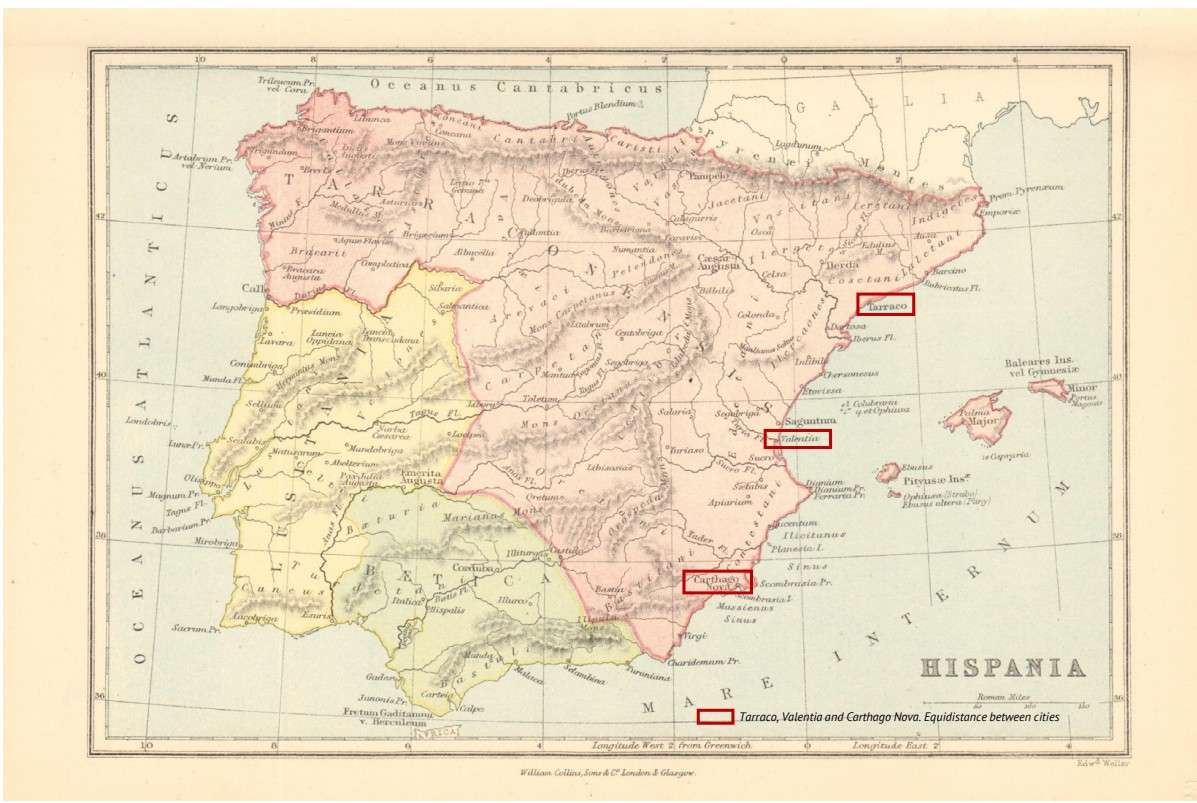

**Figure 1.** 'Hispania' Roman name for the Iberian Peninsula. Provinces. Spain and Portugal. (*Bartholomew, 1876*) [12].

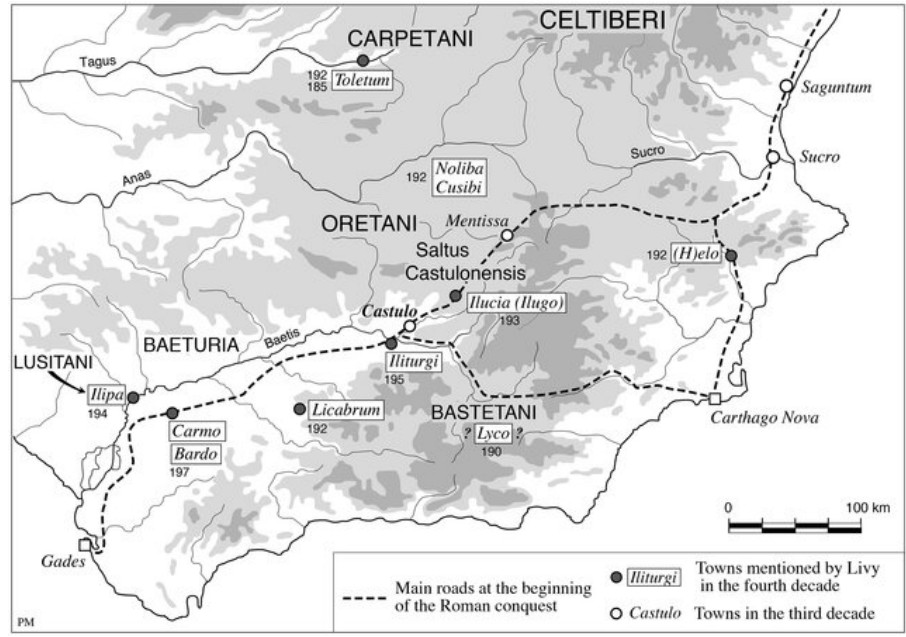

**Figure 2.** Bifurcatión of Via *Herculea*/*Via* Augusta south peninsula (*Román Turdetania, 2018*).

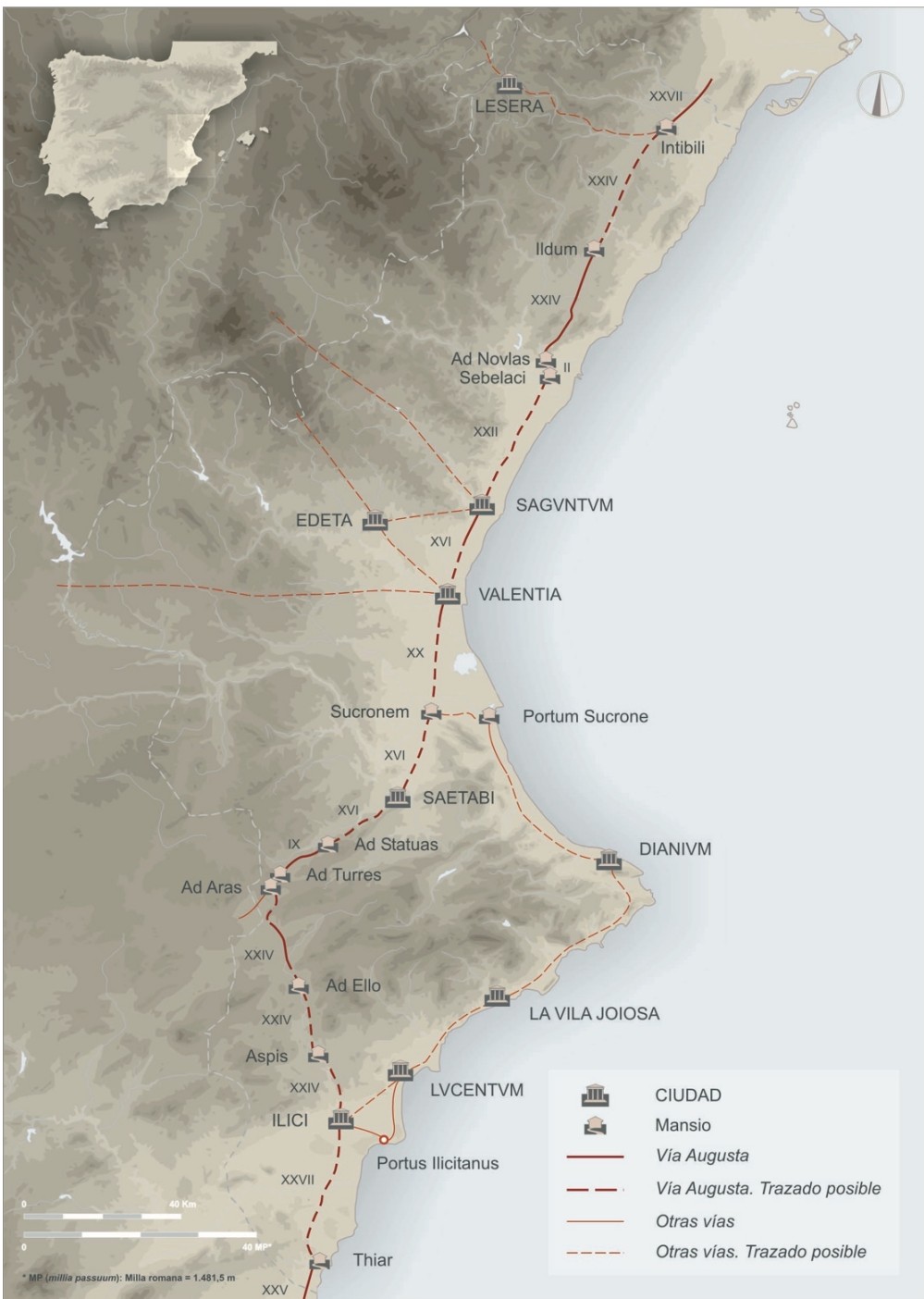

**Figure 3.** Main roads, cities, mansions and the distances between them (*Ripollés I Arasa, 2006*).

Despite the fact that the city was once built on a somewhat higher surface than the *Tyris* (Turia), the archaeological remains that have been found over the years show that the riverbed that bordered the city was lower in the Roman and Islamic periods than it is today [13]. The settlement of the first population centre was located on an elevated natural platform, well differentiated from the surrounding land and in the narrowest area of the island, embraced by the fork formed by the river in its final stretch, which, in addition to affording greater protection, also provided a point of exit.

The existence of the fluvial fork of the *Tyris* River and its proximity to the sea is confirmed by the topographical evidence of the thalweg and the bed-face and bank-spill sediment, which appear in the geoarchaeological records [13] (Figure 4). Following Albert

Ribera, one can imagine the city's environment in the years prior to its foundation as a place dominated by canals and ponds that would come to converge at certain points (Figure 5). In areas close to Mar Street and Zaragoza Square (now Reina Square), the existence of a functional riverbed in the south of the city, at least in the Roman and Islamic periods, has been demonstrated [6]. If we take into account historical references such as the Battle of Sallust, which say that the city was located to the north of the River Turia, it's foreseen **the existence of the second branch of the river fork since the army entered from the south of the island**. Figure 6 shows how the beginning of the fork of the Turia's riverbed occurs in the vicinity of the present-day botanical gardens. There, the riverbed would separate into two main branches that would form the island. The north branch, with a more abrupt layout than the current one as it does not have the parapets that currently delimit the riverbed, would have approximately been in today's current position and route. The south branch can see the streets it crosses in Figure 6b.

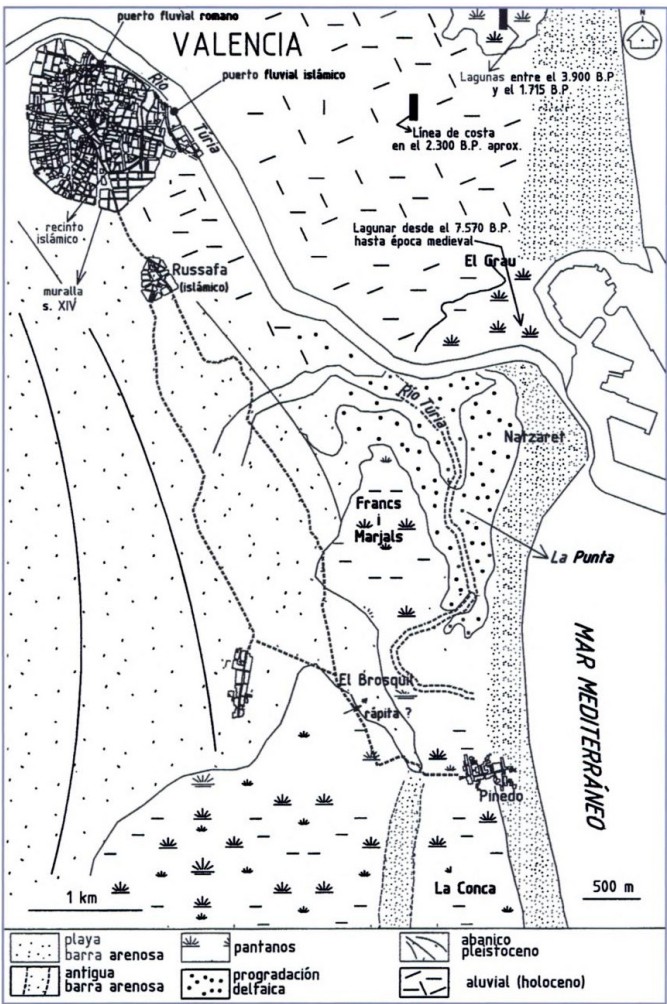

**Figure 4.** Valencian coastal geomorphological profile. The coastline of Punta d'En Silvestre (*Ruiz and Carmona, 1999*).

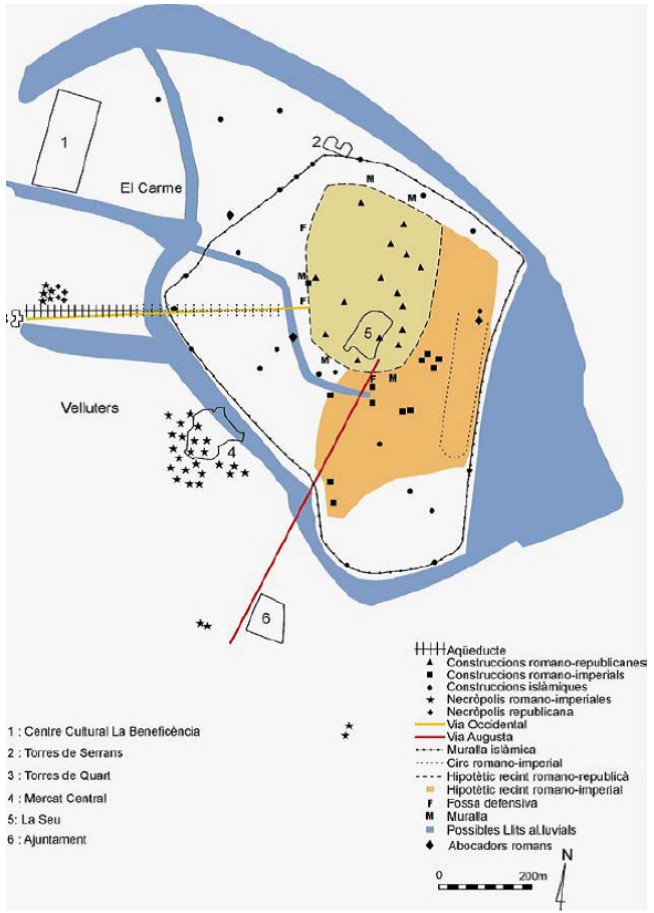

**Figure 5.** Valencia and its surroundings in the Roman period. Diseny d'Entorn (*Albert i Ribera, 2002*).

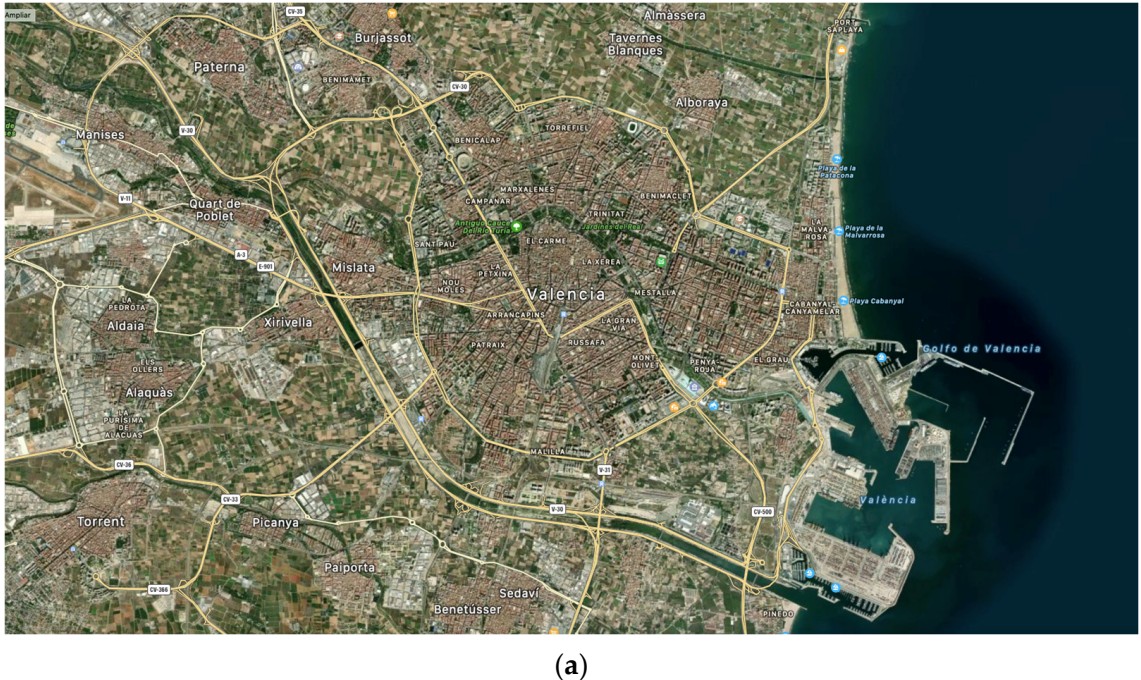

(**a**)

**Figure 6.** *Cont.*

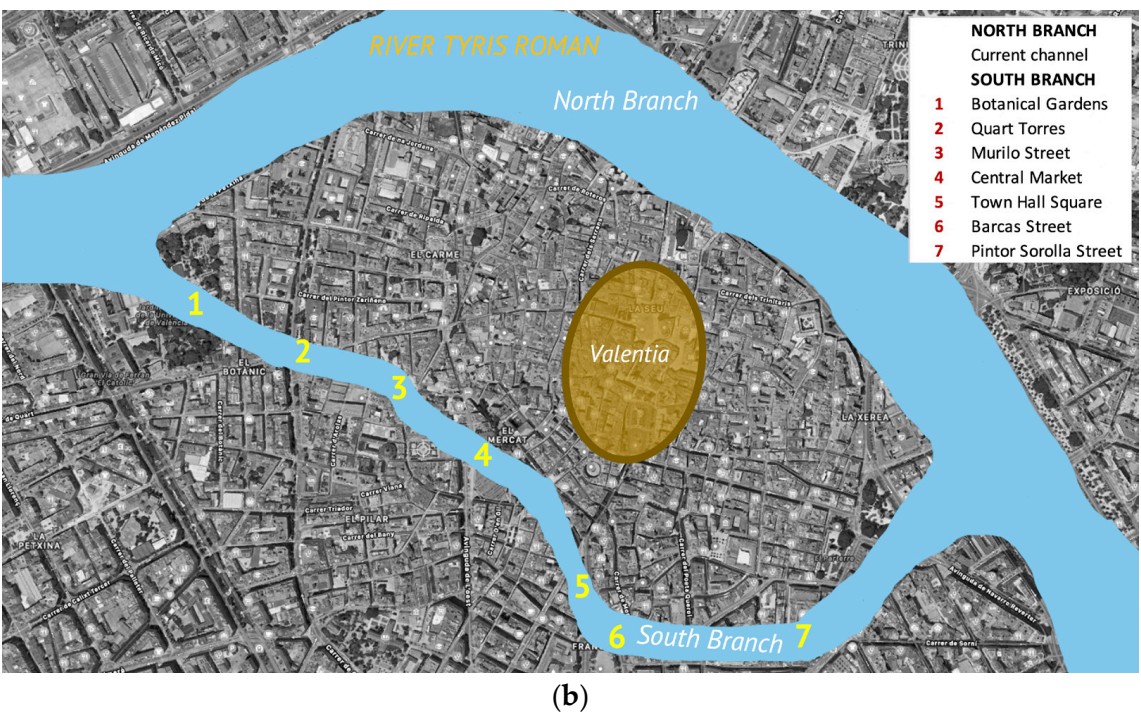

**(b)**

**Figure 6.** Location of ancient Valentia and layout of the riverbed in its last stretch. (**a**) Aerial view of the current city of Valencia. (**b**) Superimposition present-day view of the course of the River Tyris in the Roman period.

*3.2. Development of the City of Valencia in the Roman Period*

The city of Valentia boomed from the very first moment of its foundation as one of the main urban centers of the Roman province of Hispania Citerior [6]. With approximately 500 inhabitants, the first public works aimed to the delimit its area through the construction of a wall and the creation of new 'crossings.' The construction of the wall, if we follow Vitruvius's recommendations, would have had a double function: the protection of the *urbs* and the representation of the *civitatis* [6]. The creation of new crossings would be a response to the works necessary to get across the river and connect with the main roads.

The Roman city is marked by two eras:

- **Roman Republic period (138 BC to 75 BC):**

This was the era of the founding of the city, and lasted just barely more than 60 years. *Valentia*'s initial prosperity was undermined in 75 BC after its destruction by Quintus Sertorius, and it was abandoned for the next 50 years. During this period, the population, mainly made up of soldiers, enhanced and maintained the Via *Herculea* as well as the only river crossing that existed at the time, creating new crossings in the areas of interest to gain access to the island. The construction of the perimeter walls occurred during this time. These walls were defensive fortresses that delimited the cities of the era (Figure 7). According to Ribera i Lacomba [6], the city of this first walled enclosure barely exceeded 10 hectares in surface area, and its position was located practically in the middle of the island. It was further from the left branch of the river course and used the right branch of the river as a natural defense for the southern and southwestern banks of the enclosure. The urban layout of the city followed the Roman canon inherited from the Greek town planner *Hippodamus of Miletus*, with its four streets laid out on the cardinal axes: ***Cardo*** going north and south and ***Decumanus*** east and west. With regard to the population center of Valencia, the latest findings of part of this road network were discovered during the remodeling works in 2009 of the Benicarló Palace, whose archaeological work was directed by Concha Campos and is located 20 m from the Via *Augusta*. These findings attest to part of the road layout of the city that would correspond, from north to south, with the extension

of Salvador Street on its west side until reaching the right edge of the current Reina Square and west towards Arzobispo Square to Horno de los Apóstoles Street. At the crossroads of these two main roads would be the *Forum*, the nerve center of the city formed by a large square that, currently, would be located between the current Almunia Square and Reina Square. **The intersection of these two axes with the perimeter wall would generate the gates that extended the roads out onto the main roads and, therefore, to the crossings of the two branches of the river** (Figure 7).

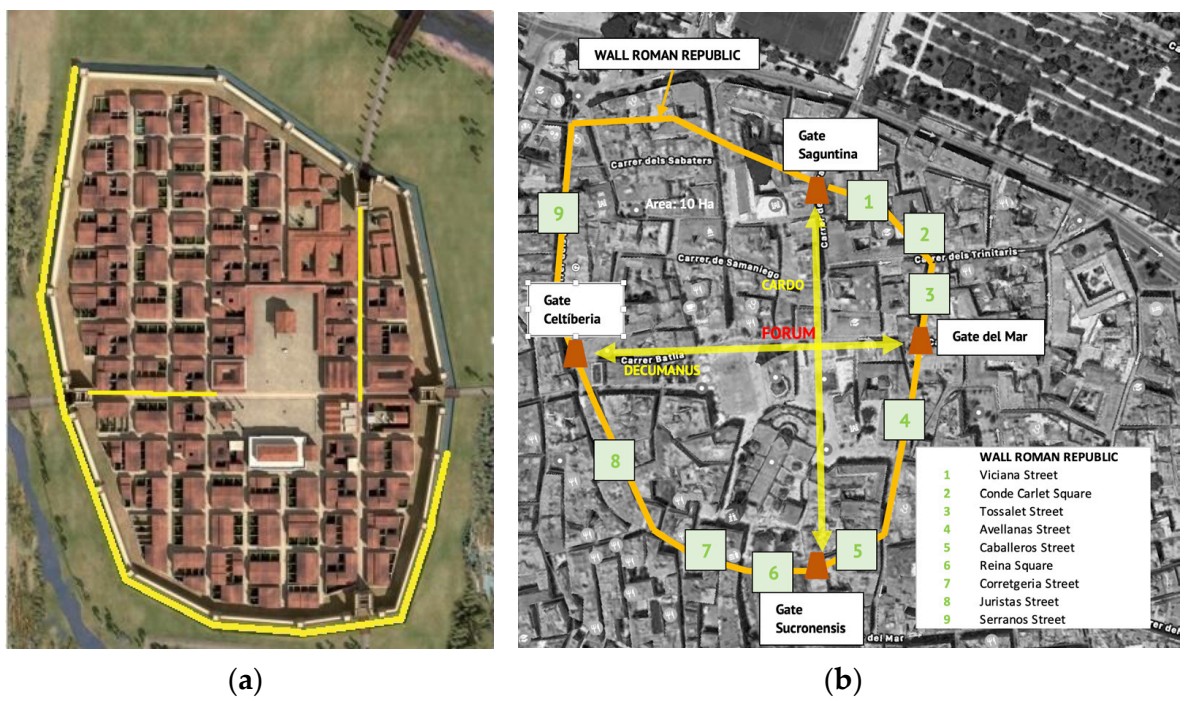

(**a**)          (**b**)

**Figure 7.** Valencia city in the Roman Republican period. Transfer to current planimetry. (**a**) Valentia city in the Roman Republican period (*Base image: ESEIESA Arquitects*). (**b**) Roman Republic city in the present-day planimetry. Preservation of some roads.

- **Roman Imperial period (ca. 25 AD to 413 AD):**

The vision of this era in the 1st century AD was approximately like that of Sanchis Guarner [11]. *Valentia* would begin to revive after the beginning of colonization by the veterans coming from the retirees of Emperor Augustus (*veterani*), the survivors of the slaughter and the descendants of the first settlers (*veteri*). This expanded the population and began the socio-economic and urban boom of the city, which culminated in its apogee with the Flavian dynasty. At the beginning of this period, the right branch of the river still existed, so the city could only expand eastwards by eliminating part of the city walls and constructing new public buildings. These included the Circus and the Port, the latter located in the area of today's Serranos Bridge (Figure 8). Its surface size would practically double that of the Republican era at around 18 hectares. During this period, work was carried out on the construction of the new Via *Augusta*, using a stretch of the existing Via *Herculea*, except in the area around the new city of *Valentia*, where the new road was consolidated to force passage through the center of the city, linking up with *Cardo*.

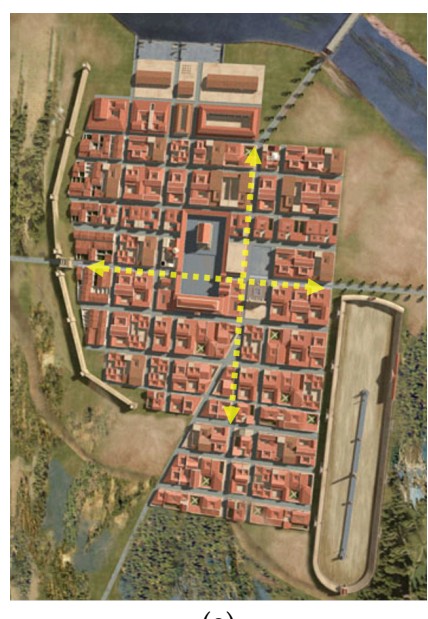
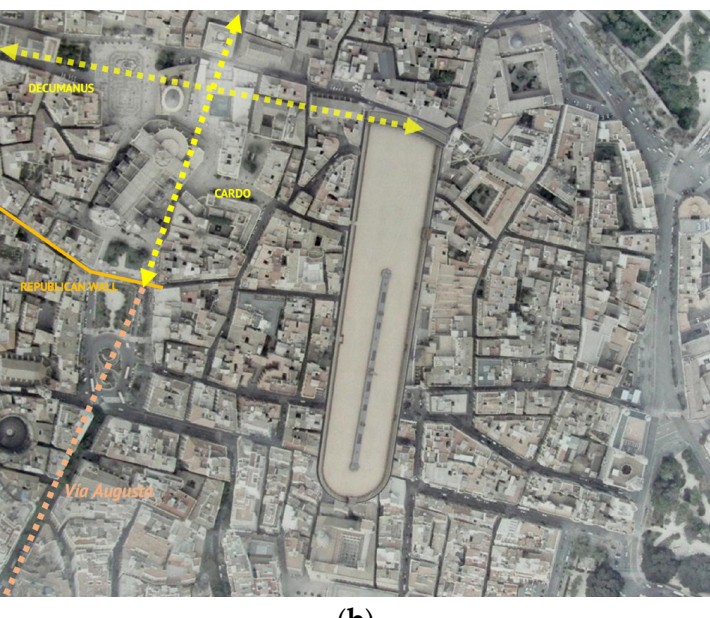

(**a**)　　　　　　　　　　　　　　　　(**b**)

**Figure 8.** Valentia city in the Roman Imperial period. Transfer to current planimetry. (**a**) Valentia city in the Roman Imperial period (*Base image: ESEIESA Arquitects*). (**b**) Roman Imperial city on present-day planimetry.

### 3.3. The Origin of the Bridges

#### 3.3.1. First Crossing: The Iberian River Crossing

In the Iberian period, the area of land corresponding to the island that would in the future surround the foundation of *Valentia* would not have been the most suitable for the Via *Herculea* to cross. The conditions of that islet may not have favoured a supposed route for the only road on the eastern side of the peninsula.

In ancient times, bridges were placed where the riverbanks were usually firmer, trying to make this type of work quick, economical and long-lasting. They would build them before the road and try to adapt the route of the road to the bridge. And as it says Durán Fuentes [14], if the road was before the bridge, it was modified. This leads us to believe that the Via *Herculea* in the vicinity of the river possibly ran around the western side of the island where the land was firmer and had the constructive and economic advantage of a single crossing. With the existence of a deeper land morphology and without limitations on its contour, the riverbed would have a greater and irregular width. In the absence of upstream dams to slow the river down, the river at that time would have had more water flowing through it at a greater speed.

The Via *Herculea*, as a means not only of travel but also of trade, triggered settlements around it. The discovery in 2008 of ceramic remains from the 3rd century BC in the area of Ruaya and Sagunto Street testify to a settlement produced by a 'trade route', reinforcing the hypothesis that the route of the *Via Herculea* would run along the west side of what, in the future, would be the first Roman city. And as Aranegui [4] pointed out at the time, "it is possible that in Ruaya Street there was a town or commercial transit enclave, an intermediate station within a route, possibly the Via *Herculea*", a theory that Albert Ribera (head of the Service of Archeology of the Valencia City Council) would ratify in his day.

This first Via *Herculea* was undoubtedly the first crossing of the River Turia before the city was founded. It would have been an Iberian crossing originally, inherited and used by the first founders as the only and fundamental crossing point of the river, fading into the background when the Via *Augusta*, which would cross the island from north to south (Figure 9), came into prominence.

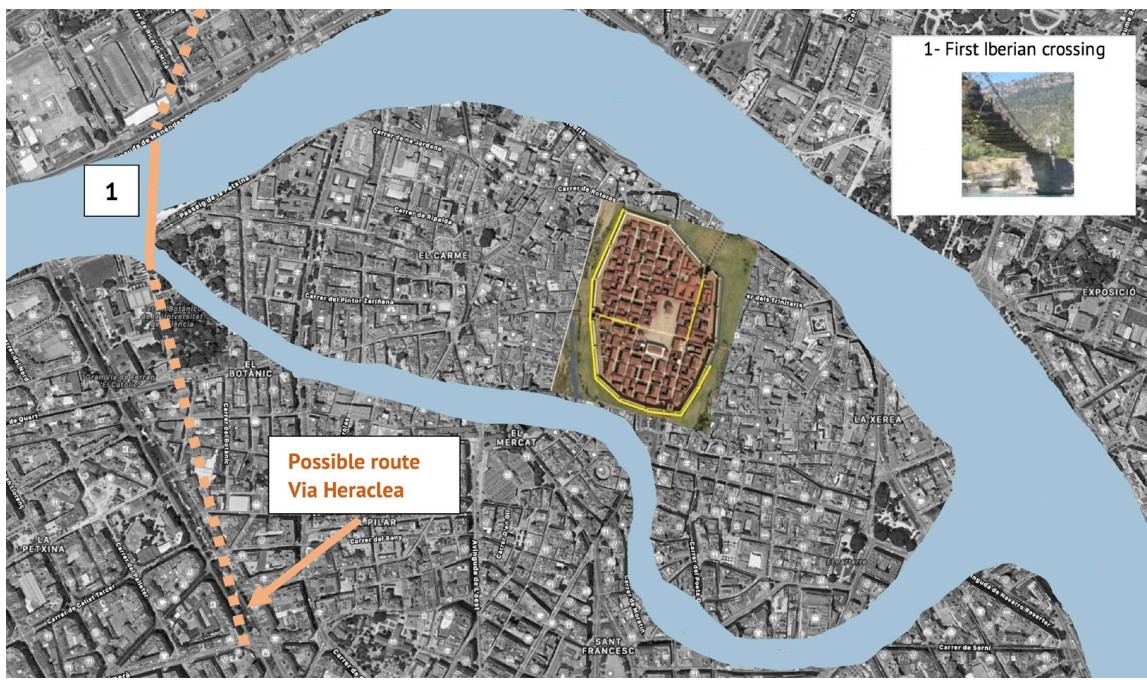

**Figure 9.** Hypothetical location first crossing.

The location at the western part of the island can be found in the archaeological finds from between 1934 and 1935 on the left bank of the current bed of the River Turia between the San José Bridge and the Campanar footbridge. These finds described by Sentandreu [15] revealed a topsoil layer about two meters thick, below which there is a layer of sand about four meters thick, beneath which there is a compact layer of vitrified gravel. Below this, gravel appeared down to the level of the present riverbed. Between the layer of topsoil and the vitrified layer, i.e., the sandy area, the remains of stone walls were found (Figure 10).

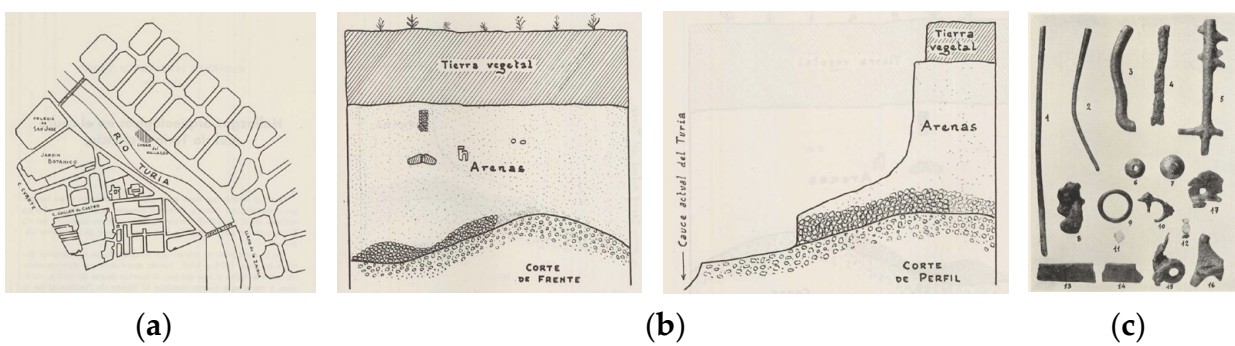

| (a) | (b) | (c) |

**Figure 10.** Ground level and elevation sketches of the location of findings of a possible Via *Herculea* crossing over the riverbed (*M. Carmen Santandreu, 1935*). (**a**) Location on the ground. (**b**) Structure elevation and cross-section. (**c**). Objects found.

The appearance of this structure is linked to a list of objects found that covered an extensive time line from the Roman Republican period through to the late Roman and Visigothic periods, with the findings due mainly to the river's drag. This does not include the vitrified layer which, as the archaeologist Sentandreu indicates, is not due to the river, although she does not know its origin and purpose. This layer of vitrified gravel was found at a depth of about six meters, sloping steeply towards the river bed and undulating in the direction of its axis (Figure 10). This would confirm the hypothesis of the location of the first crossing at the western part of the island.

The *vitrification system* was a practice used in ancient times to obtain civil engineering materials [16]. Extrapolating this to what exists in the riverbed and analyzing the level at which it was found, it could well be earth stabilization material in the form of a foundation, which, in its day, supported a perfectly delineated wall. This suggests the existence of an ancient base structure of what would be a riverbed crossing element. Although river parapets did not exist in the Iberian period, the fact that at this particular point, where the river fork seems to have started at one time, the remains of a well-defined structure were found, suggests the existence of a possible abutment supporting a crossing. It should be borne in mind that, in terms of construction methodology, at that time it was usual for this type of engineering to be carried out with stone, sandstone and material available in the area, finishing off the crossing with trunks of wood. They were works with a limited lifespan that were quickly executed and lasted for the period of use they were necessary for. As a result, with the passing of the centuries and the ravages of history, the preservation of some of the crossings remain infeasible. **Therefore, it is legitimate to think that the first step used was located in the western area of the island based on the crossing of data and discoveries made.**

3.3.2. Second Crossing: The Roman Republic Bridge

The foundation of a new city by the Romans and the subsequent consolidation of the Via *Augusta* were necessarily linked to the construction of a crossing of the right branch of the river to access the island from the south and opening the way to *Sucro* (Silla).

Extending the city road *Cardo* as part of the Via *Augusta* meant that at the new city of *Valentia* it had to cross two riverbeds: the north and south branches of the River Turia. The right crossing was affected by the extension of this road to the south through the gate built into the city wall known as Sucronensis, whose location would almost immediately link up with the river channel. This fact would have led to the imminent need for the construction of a crossing, leading to the **reason for its location**. This would therefore be the first bridge built after the foundation and probably the only bridge built in the Republican period (Figure 11).

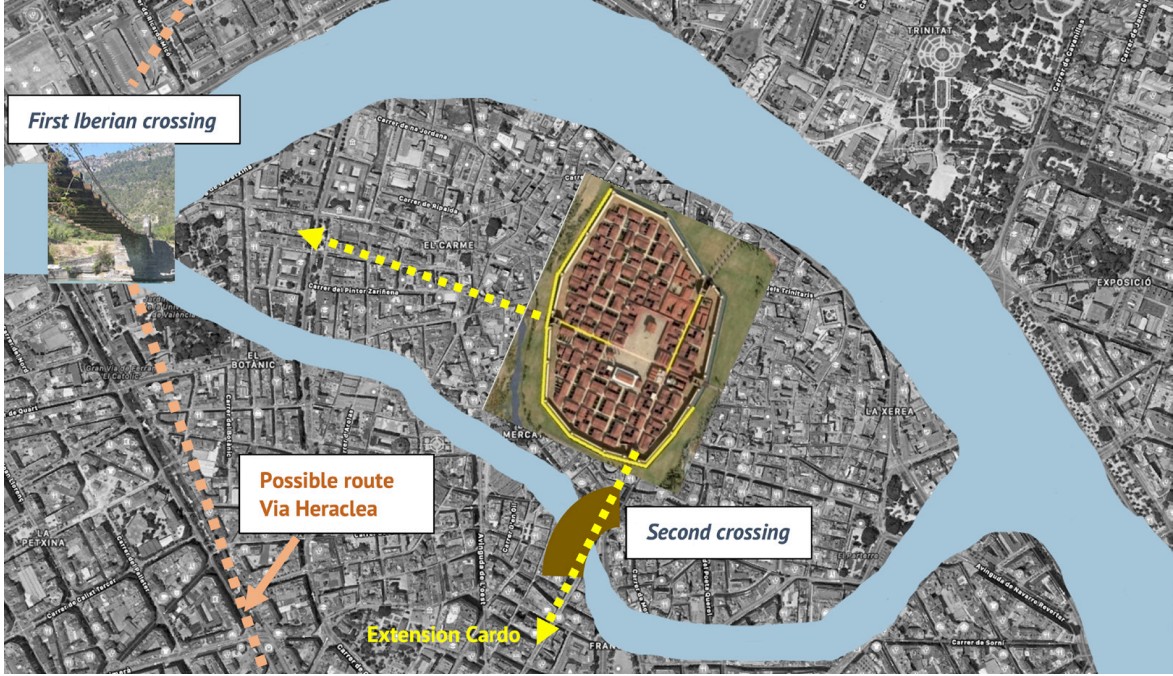

**Figure 11.** Hypothetical location second crossing.

The hypothesis of the existence of this bridge can be found in the discovery described by Ribera [6] in the Reina Square at the extension of Corretgeria Street on an imposing

construction of ashlars on a bed of natural gravel [14]. Part of the engineering works found are in line with how Romans built bridges. The *Mappae Claviculae*, documents from the 8th century AD that contain shallow Roman foundation systems, established the ratio of foundation depth to a quarter of the height of the work, recommending a base on firm ground and not to use soil mixed with stone [17], which are usually draining soil that would undermine the foundation causing the structure to collapse. Three techniques could be used to lay the foundations in the middle of a riverbed: lateral diversion of the entire course of the river, containment in the foundation area during low water with temporary dams, or the simplest procedure, which is building dry enclosures to lay ashlars on firm layers of soil (Figure 12). On some occasions, the foundations could be made with a first line of ashlars laid or with several lines that were set back from the one below, forming a slight escarpment that was advantageous to the foundations by increasing the support section and the hydraulic behaviour of the work. The latter technique seems to have been used in this first bridge.

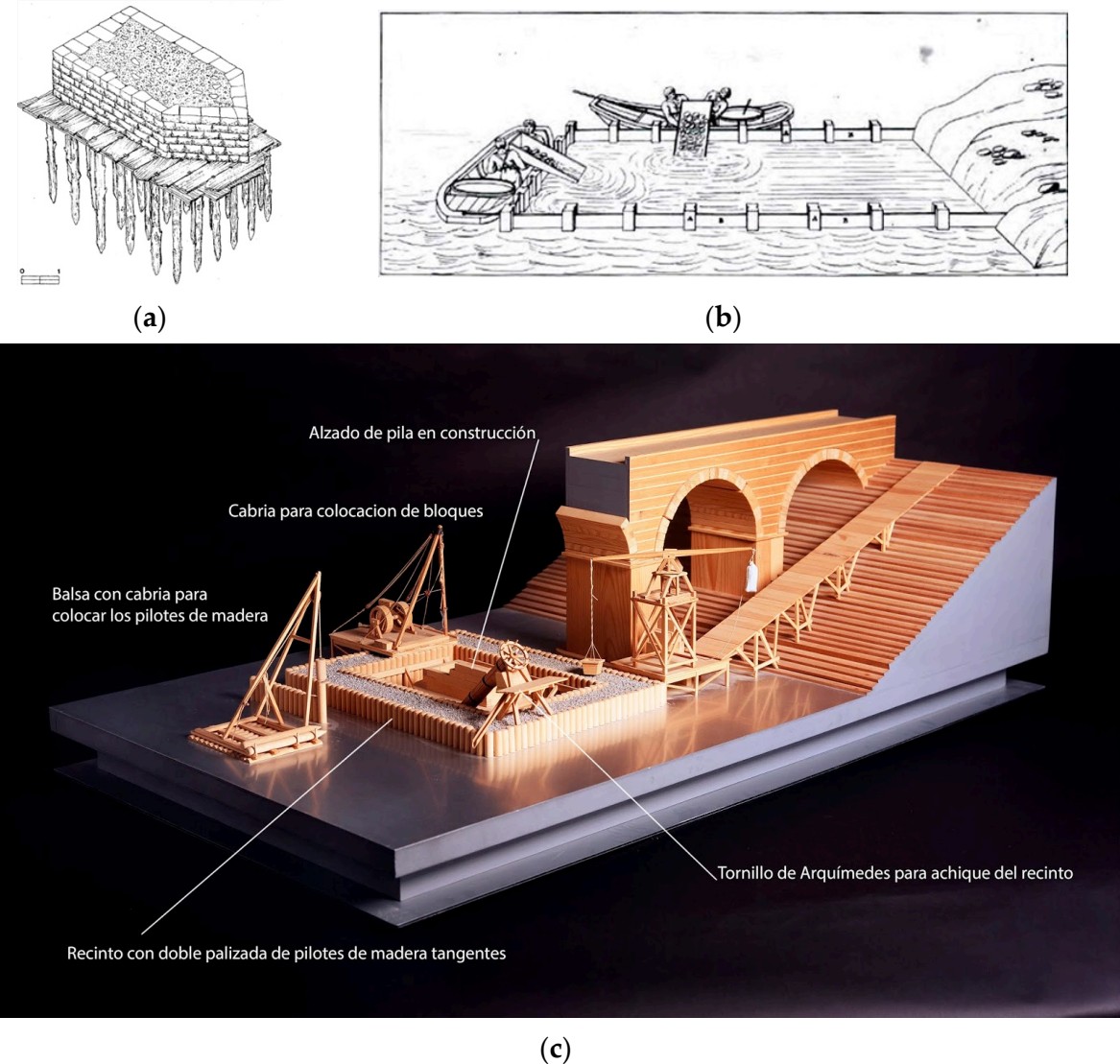

(**a**)  (**b**)

(**c**)

**Figure 12.** Execution of Roman foundations. (**a**) Detail of ashlar foundations with wooden piles on unstable ground. (**b**) Detail of construction of dry enclosure foundations. (**c**) Construction of dry enclosures with wooden piling walls for dewatering (*Model belonging to the 2005 temporary exhibition "Artifex. Roman engineering in Spain"*).

The functionality and durability of this bridge would have been conditional on the right branch of the river being stopped up or blocked. Everything suggests that this process began in the 1st century AD at a slow but steady pace, as there are now strata which appear to be alluvial in what would have been its course with abundant ceramic material staggered over time, starting from Tiberius in the middle of the 2nd century AD. This process, which would have been due to natural causes, culminates in a pavement that marks the turning point of another type of later anthropic sedimentation, alternating with episodes of flooding [14]. The fact that the fills supported the construction of the bridge would date the construction of the bridge to the Republican era.

As for the span of the bridge, it should be borne in mind that the riverbed of the right branch was much smaller than the left, so it would have had less elevation and flow. The fact that its location was practically at a tangent to the walls that bordered the city made it a weak point in the event of war, so it is more than likely that its deck was made of wood. This feature gave it versatility, allowing it to be quickly detached and protecting the city from attack.

3.3.3. Third Crossing: North Branch of the River
The Need for the Third Step

The construction of the third bridge over the north branch of the river, with greater span and flow, requires a level of resources and knowledge in line with an evolved, consolidated society in full social and economic apogee. This bridge, as part of the route of the Via *Augusta*, would be the letter of introduction of a new city in all its splendor. The Roman imperial era met these requirements. Practically doubling its population and with an economic and energy boom, they allowed the execution of several public works. These include the port in the area around the current Serrano Towers, the circus and, of course, the bridges. The extension of the *Cardo* as part of the Via *Augusta* would have ended in the vicinity of the present-day Trinity Bridge, although to date no physical evidence of the existence of a bridge in this area has been found. In fact, the lack of data in this respect is striking as it makes sense that the extension of one of the main roads leading out of the city through the Saguntina Gate would link up with the road to another of the major Roman cities of the time, Saguntum. However, there are two physical constraints that would put the execution of this work at risk (Figure 13):

- On the one hand, **the Port**, located in the middle of the Furs Square, where the current Serranos Tower is located and which would have been on the left bank of what was once the Saguntina Gate. The heyday of Valentia saw the need for building a port. The reasons included Imperial control of the Mediterranean and the important commercial boom that the city of *Valentia* would bring as the epicenter of goods traffic with the rest of the peninsula and, in particular, with its center.
- The second physical condition is the location of **the Walls** on the northwest side. The Imperial period maintained this walled area as it expanded along the east and southeast sides. This limitation, without any entrance gate, would call into question the possible construction of a crossing in the vicinity.

The possibility of the existence of a bridge in this area downstream of a port would suggest that it was navigable, and that part of its deck would have to be foldable or easily removable to allow the type of shipping that travelled at the time to pass through.

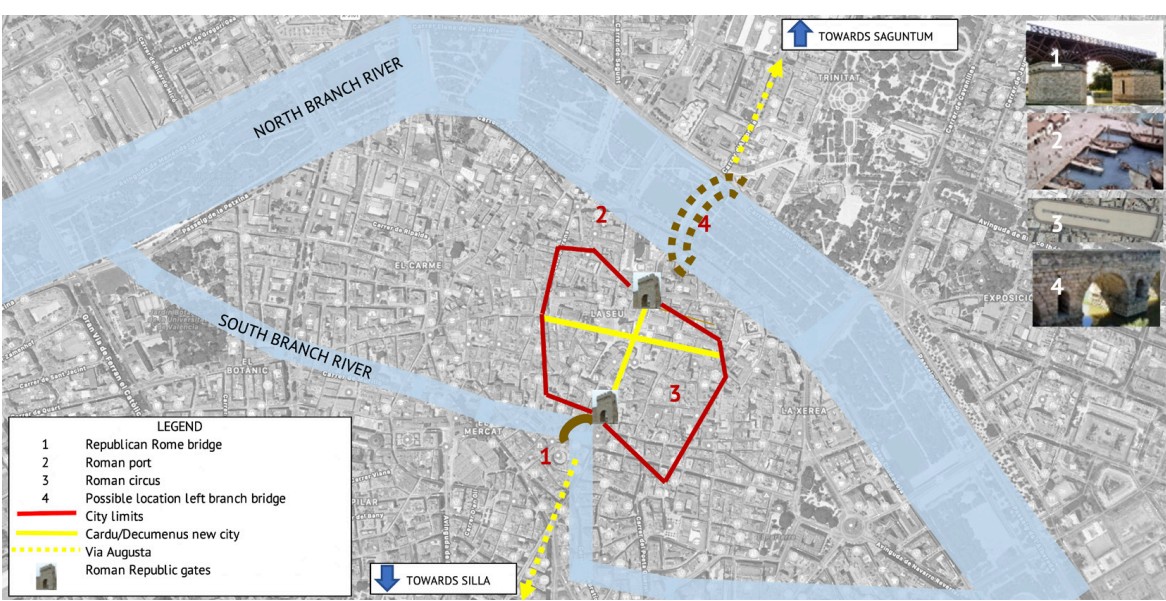

**Figure 13.** Public works in Imperial Roman Valentia.

Evidence of the Existence of the Third Step

Analyzing documentation that sheds light on the existence and possible location of this crossing, we find narratives from the Islamic period that refer to the existence of a Roman bridge on the north branch of the river, the location of which may differ from what has been speculated to date. This bridge would be **the Kantara or *Al-Qantara* Bridge**, whose name has been passed down by Arab writings, alluding to its communication with the interior and north of Spain, as well as with the Islamic *Al-Kudya* [18].

It was the first bridge built with stone in Valencian lands using Roman engineering. Although the Arabs took responsibility for its execution, Casado [19] points out that they limited themselves to maintaining and restoring it, as they did with other Roman bridges on the peninsula. Bertomeu [18] says it was built between 44 BC and 27 BC. It lasted until the flood of 1088, the oldest known to date and recorded by *Huici*, who mentions a flood of the Turia that destroyed it and the Defense Tower protecting it. *Ibn al-Kardabūs* [20] in *Loci de Abbadidis* praised the bridge, saying that there were no 'more perfect and more honourable walls' in the whole of *Al-Andalus*. The words of *Ibn al-Kardabūs* would support the hypothesis that it was built in the Imperial period, as it was characterized by its robustness, sobriety and perfection, distinguished by the typical construction style of Imperial Rome. Recent morphological and scientific analyses carried out by Xavier Bertomeu provide relevant data on its functional characteristics. Like all Roman bridges, the *Al-Qantara* Bridge was a weir bridge. This bridge stood until 1088 for several reasons:

- It could not have been independent of the river due to its location on the riverbed and the variations it underwent at that time. In periods of heavy rainfall, it would reduce its cross-section by 35%, like the majority of Roman bridges in Spain, which have disappeared precisely because they act as dams in large floods. This problem leads to an increase in the average velocity in the levee, generating eddies which can knock the piles out from under them (Figure 14). This is why Roman engineers sought to anchor their piles in solid rock.

- According to the Hydraulic Theory of the Great Flood, which assumes a return period of approximately 100 years (Figure 15) and the fact that the *Al-Qantara* Bridge lasted more than 10 centuries without being affected by any of the great floods confirms that, during its existence, the river was bifurcated in two. This would cause one of the branches to function as a bypass, reducing the damming effect of the bridge and keeping it from collapsing.

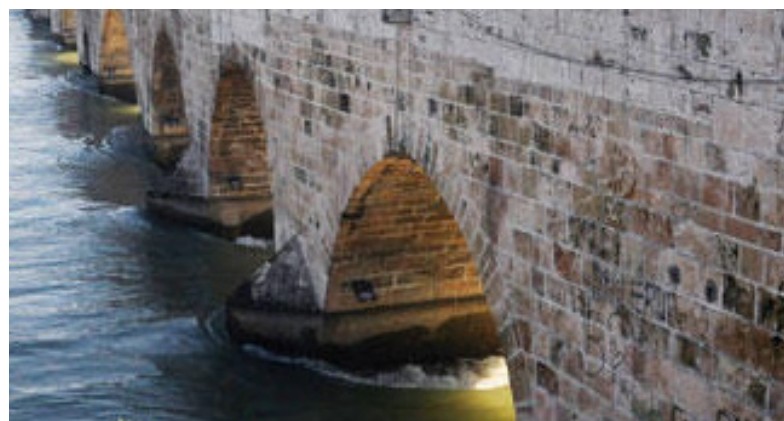

**Figure 14.** Eddy caused by increase in the average velocity in the levees.

| Period | Year | Flood estimate |
|---|---|---|
| AD | 1088 | 13th major flood (**Al-Qantara Bridge ruined**) |
| | 988 | 12th major flood |
| | 888 | 11th major flood |
| | 788 | 10th major flood |
| | 688 | 9th major flood |
| | 588 | 8th major flood |
| | 488 | 7th major flood |
| | 388 | 6th major flood |
| | 288 | 5th major flood |
| | 188 | 4th major flood |
| | 88 | 3rd major flood |
| BC | -12 | 2nd major flood |
| | -112 | 1st major flood |
| **Foundation Valencia** | -138 | None |

**Figure 15.** Estimate of floods in the city of Valentia up the collapse of the Al-Qantara bridge.

3.3.4. Location of the Al-Qantara Bridge (Third Passage of the Turia River)

Although there is evidence of the existence of the *Al-Qantara* Bridge, its location is today an enigma. Knowing the size of the city, its location and the main access routes, we will use infrastructure built in the Roman imperial era such as the Port, located in its day in the current Dels Furs Square and practically equidistant from Saguntina and Celtibera Gates. This approach generates two sub-hypotheses, the first which places the bridge upstream of the Roman port and the second, downstream of said Port.

Bridge Located Upstream of the Port

At that time, the River Turia had a navigable channel whose water flow and depth was greater than today's. Locating a bridge practically attached downstream of a port with full commercial intensity would not only obstruct its own purpose but would also invalidate its activity. The reasonable thing to do, taking into account the means available at the time, was to build bridges upstream of the port to accommodate the transit of large vessels, like at the Roman port of the Guadalquivir in Cordoba (Figure 16).

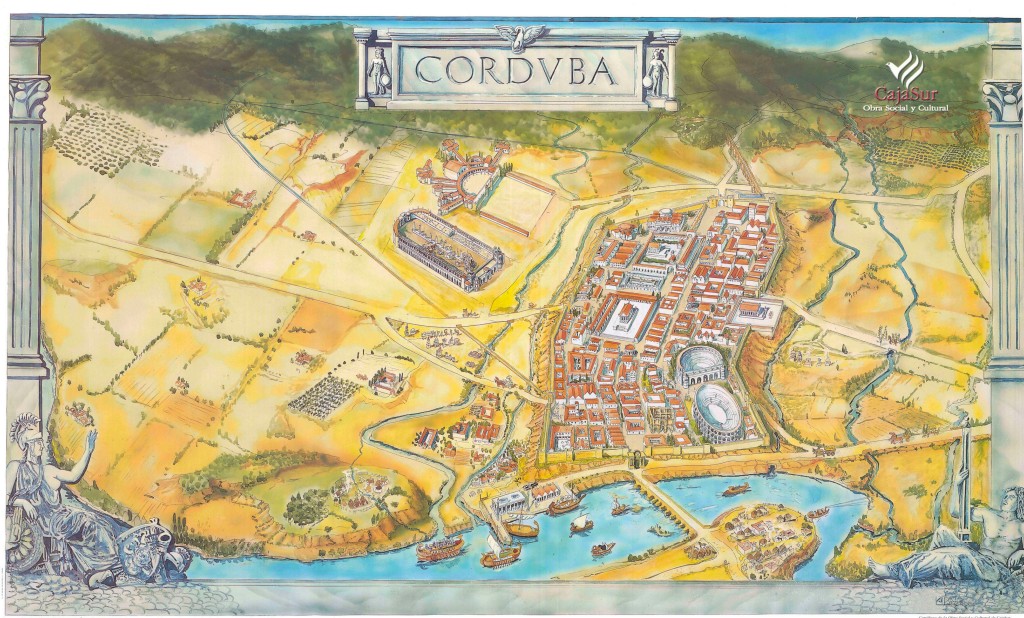

**Figure 16.** Idealization of Córdoba in Roman times. Location of the bridge in relation to the port. (*Caja Sur. Social and Cultural Work. Digital Catalog of Historical Cartography of Andalusia*).

Another fact to be taken into account that would corroborate its execution upstream of the port is the hydraulic data on its behaviour. The fact that the right branch of the riverbed acted as a spillway or bypass during heavy flooding indicates that its location must have been in the vicinity of the fork, so it would not be unreasonable to locate it near the current San José Bridge or further upstream, near what was the crossing of the Via *Herculea* (also giving meaning to the '*Hallazgos en el cauce del Turia*' or '*Findings in the Turia Riverbed*' made by Santendreu in 1935 and the link with the Serrania). On the other hand, access to the bridge could be from the Celtíbera Gate, which would provide a clearer path and sufficient distance to react to warlike confrontations. These approaches allow us to generate a plan of the possible location of the bridge that is illustrated in Figure 17.

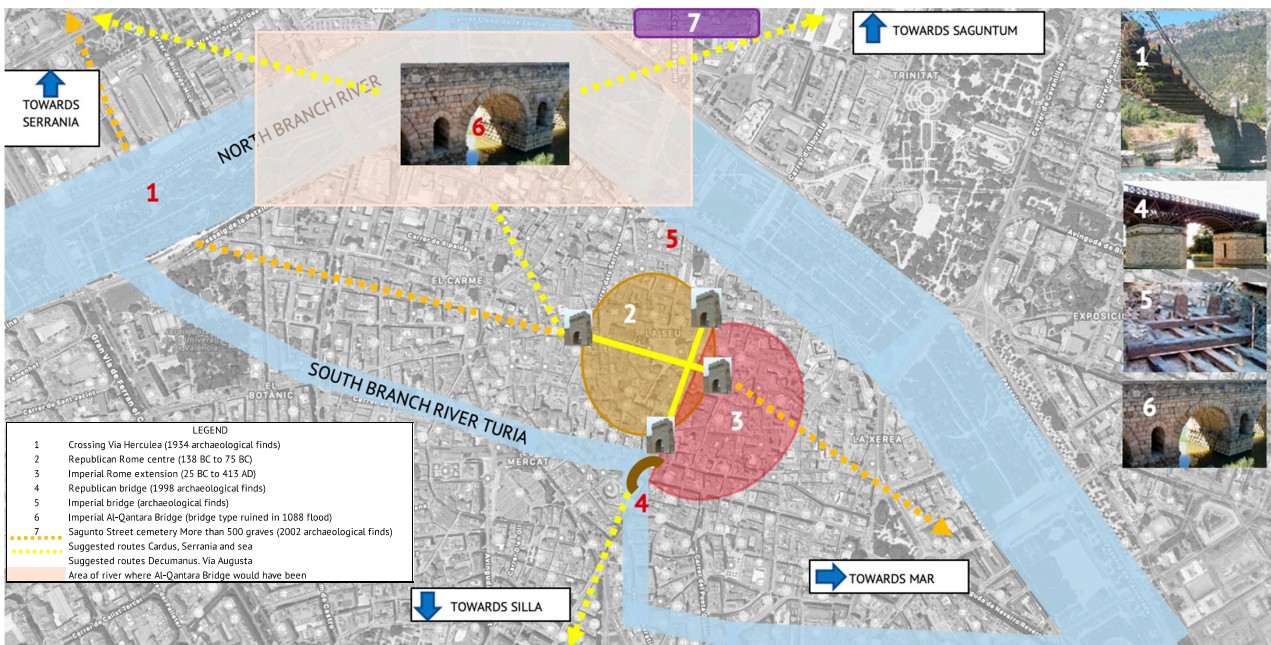

**Figure 17.** Bridge located upstream of the Port.

Within its possible location upstream of the Imperial Port, a site could also be considered at what is currently the Serranos Bridge, as some suggest this route to be an extension of the Via *Augusta*. Some writings propose that this Via ran along the current Sagunto Street, but the discovery made by Javier Máñez [3], Director of Archaeological Intervention, in 2002 on this street raises uncertainty about its route. The discovery of more than 500 burial sites across what is now Sagunto Street, occupying the block on the right, calls into question the layout of the Via *Augusta* at that point. It would have been unthinkable, keeping in mind Roman customs, to create a road across a cemetery. On the basis of these findings, the crossing of the Via *Augusta* in the first section of Sagunto Street up to its connection with Orihuela Street should be ruled out. This important piece of information would rule out a possible location of the *Al-Qantara* Bridge on the site of what is now the Serranos Bridge.

Bridge Located Downstream of the Port

Proposing the location of the Bridge behind the Port would be reinforced by four main arguments: tangent to the port must have been navigable, either through large arches or through a system of moving platforms at some point, although this is an unlikely system for a bridge of stone; located in the extension of the Cardo, it would link with the possible Via Augusta and the road to Sagunto; the *Vía Herculea* crossing would be maintained for its connection with the Serranía road, or the exit of the bridge would have been linked to a new Serranía road; finally, a much more organized and logical urban structure, typical of Roman construction standards and more so in the creation phase of the new city. The location of the bridge with this second approach can be visualized in Figure 18.

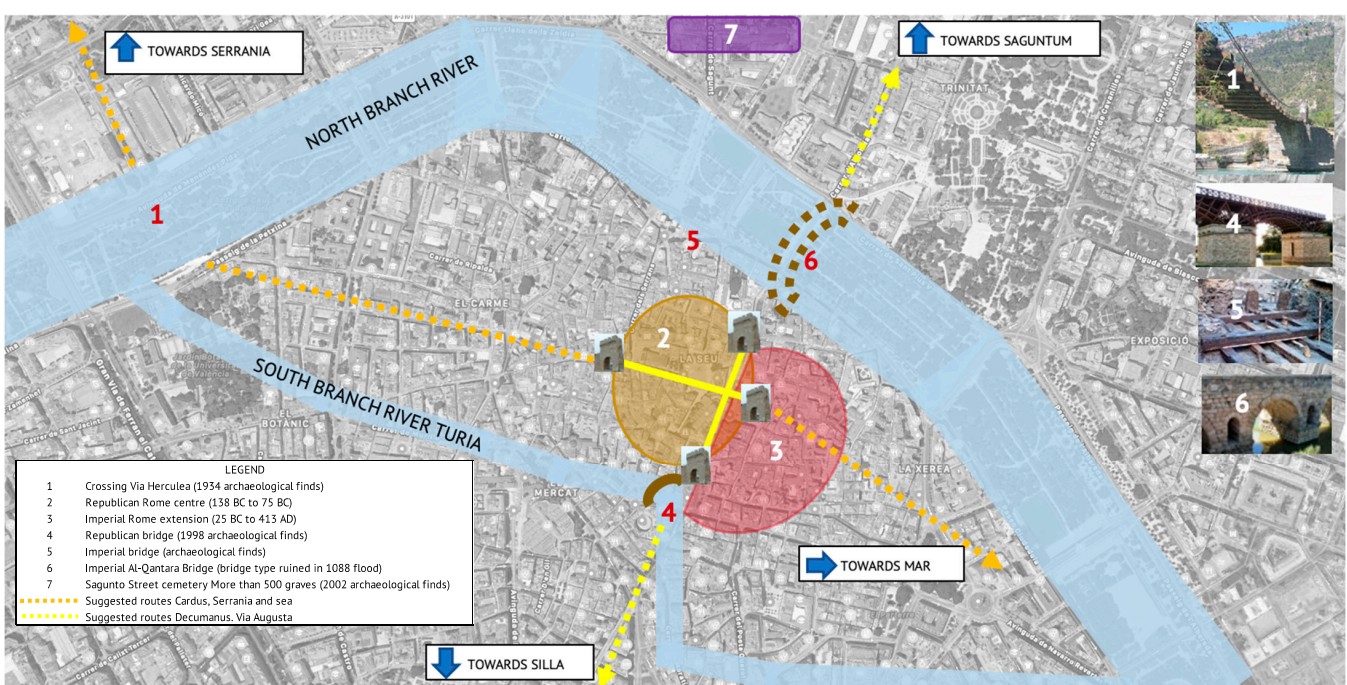

**Figure 18.** Bridge located downstream of the Port.

This second assumption approach prompts a transfer to the Islamic period on the basis of documentary references to the *Al-Qantara* Bridge. The Muslim wall was known to have consisted of seven gates, four main ones that connected the roads to the outside world and three smaller ones that served to distribute traffic among the entrances to the city. Facing the riverbed to the north was the *Bab-al-Faradj* (or *Bab-al-Warraq*) Gate, opposite which, on the other side of the riverbed, two Muslim palaces or almunias were built: Rahal and Vilanova.

In the Muslim period, the bridge inherited from the Romans served as a guide for the growth of the city on the other side of the river. This would begin with the creation

of the almunia or farm, which would later forge the formation of small neighbourhoods known as arrabales (suburbs). According to Teixidor [5], the two arrabales that emerged on the northern side of the city of Valencia, on the other side of the river, were '**bridgeheads**' (Figure 19). This period also generated a port that was far from the Roman port and located downstream in the area around the present-day Convent of Santo Domingo, as archaeological findings have shown.

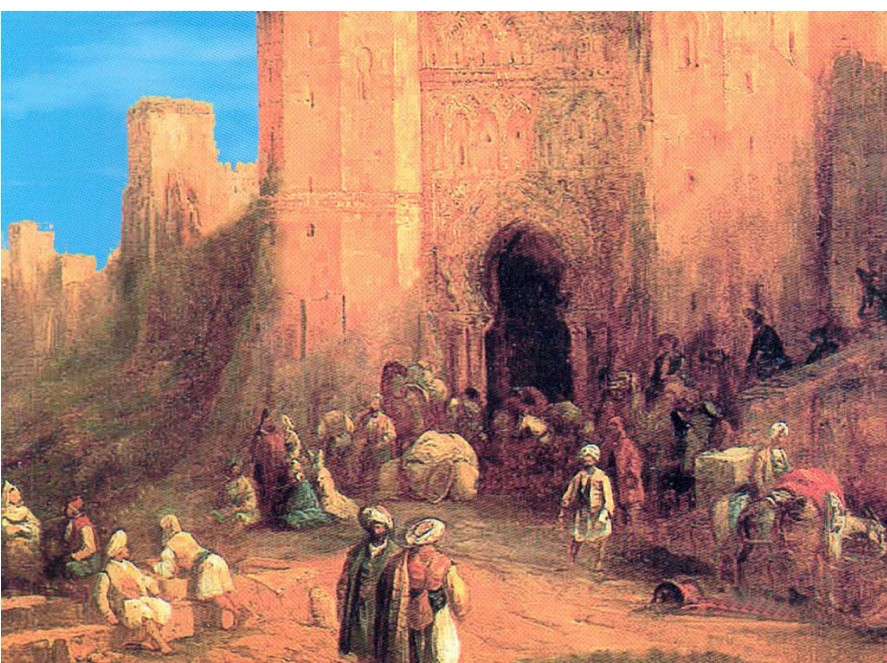

**Figure 19.** *Bab Al-Qantara* Gate facing the *Al-Qantara* Bridge (*Vicente Coscolla, 2003 "Muslim Valencia*).

This urban context gives a glimpse of the origin of two suburbs, that of Alcudia and that of Vilanoba, headed by the *Al-Qantara* Bridge, together with the creation of a new port downstream of that bridge. A detail to be taken into account that would further narrow down the location of the bridge is the fact that the subsequent Royal Palace was built on the almunia that became the Vilanova neighbourhood, as stated by various sources such as Texidor [5].

If the origin of the bridge dates back to Roman times, the fact that it is a typical Roman urban regularity linking the *Cardo* and *Decumanus* streets with the main roads would further attest to its location. The extension of these roads would mark the route of the *Al-Qantara* Bridge, placing it in the vicinity of *Cardo* Street, an extension of what is now Salvador Street on its west bank, where the last vestiges of the *Vía Augusta* have been found and in the vicinity of what is known as the Trinidad Bridge.

With regard to the evolution of the characteristics of the river basin, the process of sedimentation over the years has caused the bed to rise by several meters (hence the fact that the arches of the bridges built in the coming centuries are now smaller). This is significant, as if we go back 15 centuries, the riverbed was deeper, so the construction of bridges would be linked to a greater height, probably twice as high as today's bridges. Bertomeu [18], after analyzing and calculating the morphology of the *Al-Qantara* Bridge, argued that it would have had 10 arches with a span of 9.2–9.8 m and a span/width ratio of 2.9 m. The two largest arches would have been located in the middle of the bridge as they were the entrance to the river and would enhance its aesthetics. With this morphology and taking into account the size of the vessels in Roman times (until the 16th century the maximum size of a galley was 140 feet in length, 20 feet in breadth and with a foot draught), it would not block the passage of ships coming off the Mediterranean.

Resolution of the Possible Location of Al-Qantara Bridge

The investigations carried out on the two assumptions allow a comparison to be made that specifies the possible location of the Bridge. Regarding the first assumption, no written or graphic evidence has been found that locates the bridge upstream of the Roman Port. Locating it in that area would place it too far from the population center to be newly built and probably too close to the first Iberian pass to be urban-functional. It should be remembered that the possibility that it was in the vicinity of the Serranos Bridge prior to the Roman Port is dissipated by the discarding of the route of the Via *Augusta* in the first section of Sagunto Street after the discoveries of Mañez [3].

As for the second assumption, it would be reinforced by the fact that there is written evidence in Islamic times confirming a population center on the other side of the river as a "bridgehead", downstream from the Port. This fact, together with the findings of part of the Via *Augusta* on the inner margin of Salvador Street, would reinforce the hypothesis of its location in that section. At the same time, taking into account the dimensions of the bridge arch obtained by Bertomeu [18], it would resolve the only doubt that would remain regarding the functionality, since it would facilitate the transit of galleys and, therefore, its functionality downstream of the Roman Port.

These data encourage us to rule out the first assumption of locating the bridge upstream of the port despite its better hydraulic performance. Therefore, it is reasonable to consider the location of the *Al-Qantara* Bridge downstream of the Roman bridge and in the vicinity of the current Trinity Bridge as an extension of the recently banished Via *Augusta*.

Based on Figure 18, the location of the *Al-Qantara* Bridge would correspond to the extension of the Cardo on its northern side as part of the route of the Via *Augusta* and currently coincide with the surroundings of the Trinity Bridge.

The proposed development takes shape after the recent archaeological findings derived from the replacement works of the drinking water network of the last century, which circulated along the edge of the river bed between the Serrano Towers and the current Real Bridge. These have allowed us to discover about 20 m of Muslim wall and the *Ali-Bufat* tower next to the Temple Palace, as well as the old Trinity Gate built with reused ashlars from Roman times [21].

*3.4. Durability of Bridges up to the 11th Century*

3.4.1. Evidence of the Blocking up and Disappearance of the Left Branch of the River Turia

An important event in the evolution of the city of Valencia was the fact that **it stopped belonging to an island** after a blockage of the river's south branch. *Valentia* was the epicenter of an island integrated into the peninsula with a direct connection to the sea and strategically located between the two main ports of the time. However, today the city of Valencia is not an island, as it is completely integrated into the Iberian Peninsula and is bordered to the north by a single riverbed of what was once the Turia River.

As mentioned above, the first record of a major flood in the city of Valencia was in 1088, and the fact that the *Al-Qantara* Bridge **lasted more than 1000 years without collapsing** was due to the existence of the second branch that acted as a spillway. It is well known that the stone bridges built by the Romans were weir bridges. In times of high water they could reduce the riverbed by up to a third, which caused overflows and flooding of the surrounding area. In some cases, this provoked the collapse of the bridge itself.

The *Al-Qantara* Bridge was not damaged until it collapsed, probably because the second branch of the river acted as a spillway. This also meant the bridge did not have to withstand heavy loads and collapse. On the other hand, major overflows were created, leading to flooding. The question remains as to **when and why this second course was blocked**?

According to Pertegás, the consul *Decimus Junius Brutus*, during the period of splendour of Roman Valentia, to disarm Viriathus's warriors gave them '*agros oppidium que quod Valentia vocatum est*', blocking up the second channel to create land and obtain greater benefits and therefore widening the current channel. According to Nicolau Primitiu's 1920

theory, this second branch only had water in floods. This fact would have slowly blocked it up, affirming its clogging at a time prior to the Muslim invasion.

However, it was Bertomeu who came closest to the reality of what happened on the basis of what he called 'a mathematical scientism of hydraulic engineering and bridge construction'.

*Al-Qantara* **Bridge**: This was the first Roman stone bridge of any significance, which lasted up to 1088. It is considered a weir bridge (retains 35% of the flow).

**Hydraulic engineering**: Every 100 years there is a major flood (many of the floods recorded to date are not classified as major, like the one in 1957 that devastated the city of Valencia).

**Roman bridges in the Levant**: There were many at the time. The torrential floods in the eastern part of the peninsula means most of them have disappeared. The fact that the *Al-Qantara* Bridge did not collapse was due to the spillover effect of the second riverbed.

**Size of the right branch of the riverbed acting as a spillway:**
Data:

o   *Q = SxV The flow of a river is the product of its cross-section and the velocity of the water.*
o   *Bridge weir retains 1/3 flow*
o   *Flow passing through bridge weir: 2/3*
o   *Flow rate through left branch ≥ 1/3*

$$Qt = Q1 + Q2$$

$$Q1 = V1 \times S1$$

$$Q2 = V2 \times S2$$

$$S2 = 1/3 \times S1$$

where:
*Qt: Total upstream flow before fork*
*Q1: Left branch flow*
*Q2: Right branch flow*
*V1: Water speed left branch*
*V2: Water speed right branch*

Evidence of the capacity/size of the second branch of the river: to prevent the weir bridge from collapsing, at least 1/3 of the flow held up must be absorbed by an overflow channel. As the cross-section is proportional to the flow rate, in the particular case of the *Al-Qantara* Bridge, the cross-section of the right branch that acted as a bypass was at least 1/3 of the cross-section of the left branch.

Evidence of sedimentation: According to fluid dynamics, the south branch would see an increase in its usual flow rate as it received the 'overflow' water from the left branch due to the location of the bridge. The flow in this stretch must have increased at a slower speed and sediment quantity that favoured sediment deposit. The force of the water would tend to push the torrents towards the concave part of the riverbed, forming a bend and eventually strangling it, clogging up of the river's right branch.

It should be borne in mind that when the water comes down off the mountains and reaches the main river valley or the coastal plain, there is a sharp decrease in the gradient which also reduces its capacity to carry sediment. This process in the Mediterranean area is greater than in other parts of Spain due to the lack of water. This is the reason why most of the Roman bridges in this area disappeared or were buried, as demonstrated by Fernández Casado [19].

An example of this phenomenon can be found in the River Nalón migrating at Olloniego when a meander was formed and sediments were dumped, blocking it up some 300 years ago, rendering the bridge built there unusable (Figure 20).

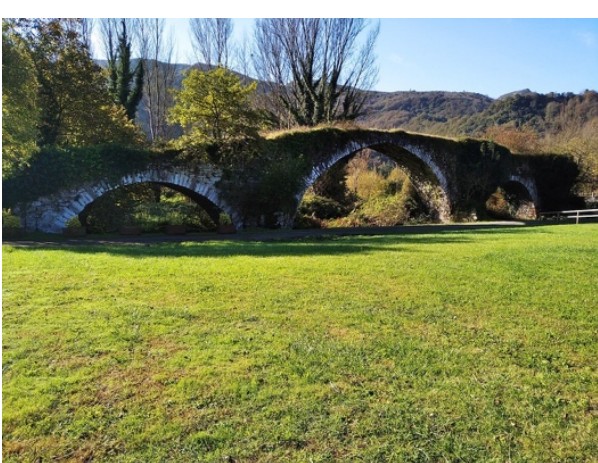

**Figure 20.** Olloniego Bridge.

We know this until the second century AD. Much of the sedimentation that occurred was due to natural causes, as explained by Ribera [6] when he points out that the southern arm of the river was in a drought phase as demonstrated by the sediment strata that delimited the date of the blinding around the 11th century. This important data is linked to fluid dynamics since in periods of heavy flooding, the southern arm received excess water from the north branch, increasing its flow and decreasing its velocity which favored sedimentation. Taking into account the calculation that shows that the *Al-Qantara* Bridge succumbs to the overflow of water from the north branch after more than 10 centuries standing, reinforces the fact that the bypass function of the southern arm ceases to exist when it relieves the 30% flow that the Roman bridge needed to remain standing. If we add history to archeology and hydraulics, we return to Huici's writings in which he confirms the collapse of the stone bridge due to the strong avenue that occurred in the year 1088. The point is that the right branch of the River Turia began to lose its hydraulic capacity and therefore ceased to exist. It was at that precise moment in which the city of Valencia stopped belonging to an island, marking a before and after not only of urban planning, but with the arrival since then of strong avenues that have overflowed the river bed, lashing the city for centuries.

3.4.2. Period Durability of Bridges

The different periods and settlements that the city of Valencia had seen up to the 11th century created bridges with a useful life that depended above all on the construction processes of the time, the needs of the different cultures and the geomorphological evolution of its riverbeds (Table 1).

**Table 1.** Durability of bridges on the 11th century (own elaboration).

| 4th Century BC to 2nd Century BC | 1st Century BC to 2nd Century AD | 5rd Century AD to 8th Century AD | 8th Century AD to 10th Century AD |
|---|---|---|---|
| Via Heraclea bridge | | | |
| | Roman Republic bridge | | |
| | | Roman Empire bridge (Al-Qantara Bridge) | |
| IBERIAN PERIOD | ROMAN PERIOD | VISIGOTHIC PERIOD | MUSLIM PERIOD |

o    Via *Herculea* Bridge: The bridge was built in the Iberian period with a deck of wooden trunks on wall piles built with stone. It would have been located upstream in the area

around the fork of the river and was in use until the 1st century BC, the beginning of the Roman period.

o    Roman Republic Bridge: It was built in Roman times and dates back to the Republican era. Its piles were made of ashlar, most probably supported by a wooden board. There is evidence of this bridge on the right bank of the river at the Reina Square. It fell at the end of the 2nd century AD before the imminent conquest of the Islamic empire.

o    Al-Qantara Bridge: The city's first stone bridge and one of the most important in the east of the peninsula. It was built in the 2nd century AD by the Roman Empire, located on the left bank of the River Turia, and was inherited by the Muslims who maintained it. It collapsed in the great flood of 1088.

### *3.5. Influence of Bridges on the City's Development*

3.5.1. Influence of the First Crossing

The first city project was during Republican Rome, when the Via *Herculea* continued to be the main road while work was being completed on the new Via *Augusta*. The basic job of the population, one of military origin with an advanced knowledge of engineering, was to build a city which would involve the creation of civil works. The location was decided on by the Romans and was adapted to satisfy the needs and provide relief for its inhabitants, making it possible for them to find the elements necessary to meet the needs for which it was created on site, as Teixidor [5] comments. The new *Valentia* was placed in the middle of the island due to the need for more security than the river alone could guarantee, the existence of a means of communication that the existing river crossing provided and the presence of an agricultural wealth that could feed the population. After initial establishment, one of the first activities was to build the defensive wall, following the typical and orderly Roman urban layout.

**The first crossing inherited** and probably renovated by the Romans provided them, in addition to accessibility and communication with the rest of the main cities, with an entry route, together with river navigation, and with the building materials (stone and wood) necessary to construct the new city (Figure 21). Due to the works needing to be carried out and Via *Augusta* under construction, the first crossing was used practically throughout the entire Republican period and was an important element in the construction of the second crossing (the first bridge built by the Romans on the island).

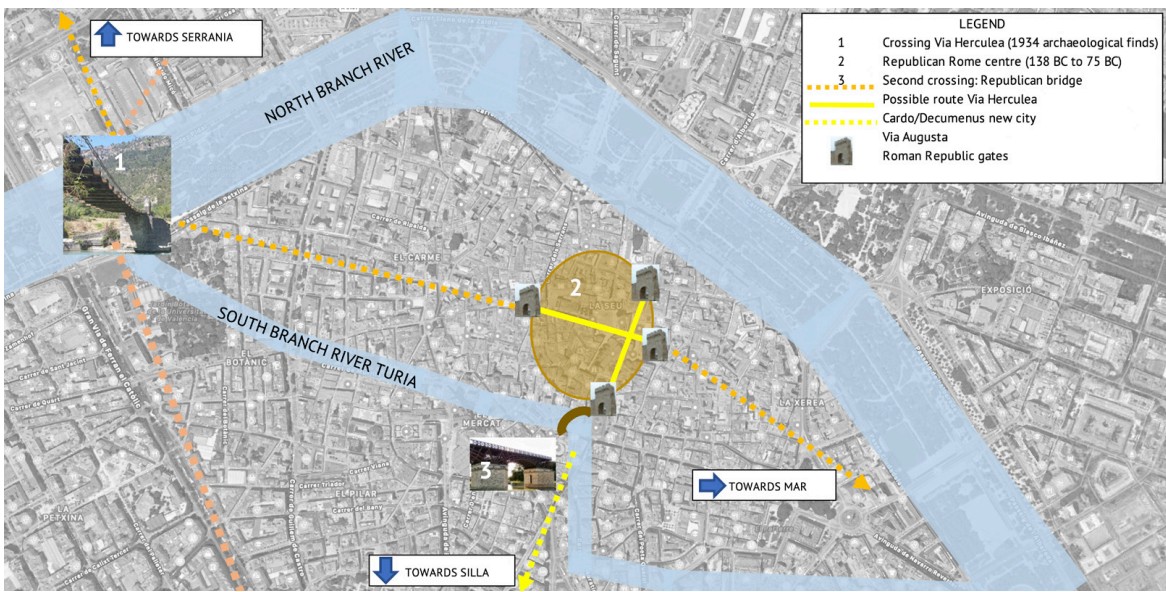

**Figure 21.** First and second crossing. Connection with communication routes.

### 3.5.2. Influence of the Second Crossing

In the case of Valentia, part of the Via *Herculea* would be used to cross the city, necessary for any city that aims to be relevant. As Teixidor [5] points out, this helped Valentia become an important nerve center in the east of the peninsula, administratively controlling the so-called Iberian Devotio or fealty of the surrounding area together with fertile and profitable lands. This second crossing, besides being necessary, had a peculiar relevance: **the consecration of the new settlement as a city**. It marked the route of the new Via *Augusta* just outside Valentia in its connection with the south of the peninsula, facilitating not only communication and trade, but also the attraction that the creation of a new city could mean for a recent Roman society that was establishing itself on the peninsula. For centuries it would be the transition piece in the main communication route from the east of the peninsula to the cities on its southern side (Figure 21) and, although the conflicts of war and the beatings endured by the riverbed affected part of its structure, it would be a fundamental construction for the city's connections and was continuously maintained by the Roman population.

### 3.5.3. Influence of the Third Crossing

After approximately 50 years of being abandoned after the Roman Republican era, *Valentia* was re-established in the time of Emperor Augustus as an Imperial city. During the Flavian dynasty it underwent significant urban growth, becoming one of the most influential cities. The city grew to the east and southeast which led to the existing city wall being relocated to that area. The new area that housed the city would be around 18 ha with a population of around 2000. This growth was linked to the construction of important buildings such as the forum and important public works such as the port, the circus (with a capacity of 10,000 spectators) and the first stone bridge. For practical purposes, the city occupied the same area and there was no need to build more bridges.

In the Islamic period (from 718 AD to 1238 AD, lasting 520 years), the city of Valencia, which the Muslims renamed Balansiya, entered a long period of Islamic rule. The main changes took place in the 10th century when the city experienced intense population growth leading to a major urban expansion. The Roman wall was demolished and a new walled enclosure was built, occupying approximately 50 hectares and housing a population of 15,000. The extension, which is much more developed in the east and south of the city, managed for the first time to directly surround the bed of the River Turia on its northern side. This new wall communicated with the exterior by means of seven gates known in Arabic as '*babs*', two of them directly facing the River Turia. During this period, the right branch of the River Turia disappeared completely. Muslim cities, with a strong religious character, are characterized by a lack of regularization, with a redistribution of streets in accordance with the typical physiognomy of the *taifas* or independent Muslim principalities: narrow, convoluted dead-ends (*azucats*). During this period, the city **survived with only one bridge.**

Below, Table 2 shows the condition of the first bridges in the first periods of the city of Valencia.

**Table 2.** The scope of the first bridges (*own elaboration*).

| Period | Stage | Area | Population | Bridges/ Crossings | Type | Bridge Origin |
|---|---|---|---|---|---|---|
| 138 BC–75 AD | Roman Republic | 10 Ha | 500 ppl | 2 | Mix of ashlar and wooden deck | Iberian and Roman |
| 125 AD–413 AD | Roman Empire | 18 Ha | 2000 ppl | 2 | Mix of ashlar wooden deck Stone, imperial style | Roman |
| 413 AD–718 AD | Visigoth | 18 Ha | 2500 ppl | 2 | Mix of ashlar wooden deck Stone, imperial style | Roman |
| 718 AD–1.238 AD | Islamic | 50 Ha | 15000 ppl | 1 | Stone, imperial style | Roman |

## 4. Conclusions

The building of a bridge marks the direction of a road and in the case of Valencia, without bridges there would be no city. The creation of cities in the middle of islands requires a prior analysis of the viability and connectivity necessary for the location of a new city. Valencia is no exception.

Much research has been conducted on the origin and development of the city of Valencia, however there is still uncertainty about the way in which the main roads in the surrounding area were linked to the city center. The research generated in this article has sought to shed light on the first bridges of the city and how they may have influenced, together with the first settlements up to the Islamic era, its urban development. This research was driven by the interest in knowing which ones were built and why, given the lack of documentation attesting to their existence. These bridges are essential for movement and connection in any important city and have evolutionary causes that, in the specific case of Valencia, allow us to gain a first glimpse of the possible urban expansion and of the main communication routes connecting with the rest of the territory.

The fact that it was a city founded by the Romans has made it possible to start from a systematic way of doing and living, which has shown how the founders made their first contact with the island. Arriving is preceded by a 'being able to arrive' and this, in the particular case of Valencia, derives from the existence of a previous road with an original bridge that was already used by the Romans over the Via *Heuráclea*. It has been demonstrated that this bridge would have been located in the western vicinity of the island next to the fork in the riverbed.

Once the city was founded, the settlement of a Roman civilization with a military spirit and extensive engineering knowledge led to the almost immediate construction of a new road through the center of the new city. This required a city like Valencia, originally located on an island, to have the elements to cross a double riverbed.

We have managed to discover the number of bridges that were built up to the 11th century, their location, typology, how long they lasted and, very importantly, the civilization that built them, highlighting a fundamental element of any road when faced with a geographical feature. The hitherto unpublished confirmation of the existence of three bridges as the main crossing points up to the 11th century, without ruling out the possible presence of crossings to facilitate passage at points of interest, is important. Of these three bridges, the first stone bridge built by the Romans in the Imperial era in Valencia, known by the Muslims as *Al-Qantara*, is of particular relevance. We have been able to locate and prove its hitherto unknown location in the vicinity of today's Trinity Bridge, downstream from what was once the old Roman port. The fact that in the impasse between the Roman and Muslim eras Valencia ceased to be an island, affecting the bridge that once linked the south branch of the riverbed, allows us to understand the way in which the city developed and the use that was made of the land. The blocking up of this branch changed the river's hydraulic behaviour, leading to the subsequent catastrophes that would ravage the city for centuries.

Extrapolating to all those cities with waterways, it seems logical to think that they should like to know the origins of their first bridges. This not only makes the emergence of the city more comprehensible, but also offers a glimpse of the 'bridge' as an essential part of the main guidelines for building processes, as a better knowledge of the origin of bridges leads to a better understanding of urban evolution.

To conclude, it is worth highlighting the use of a mixed methodology in which different disciplines such as archaeology, history and engineering converge, are decisive for obtaining a spatial and complete vision of the motifs and processes involved in the building of the first bridges to ultimately better understand the city.

**Author Contributions:** Methodology, M.-M.D.-A.; Validation, M.-M.D.-A., E.G., J.-S.P.-J. and J.L.M.-G.; Formal analysis, M.-M.D.-A., E.G. and J.-S.P.-J.; Investigation, M.-M.D.-A.; Resources, M.-M.D.-A.; Writing–original draft, M.-M.D.-A.; Visualization, M.-M.D.-A., E.G., J.-S.P.-J. and J.L.M.-G.; Supervision, M.-M.D.-A., E.G. and J.-S.P.-J.; Project administration, M.-M.D.-A., E.G. and J.-S.P.-J. All authors have read and agreed to the published version of the manuscript.

**Funding:** This research was funded by Universitat Politècnica de València [PAID 12-22].

**Data Availability Statement:** The data presented in this study are available on request from the corresponding author.

**Conflicts of Interest:** The authors declare no conflict of interest.

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
