# Peer review of "Bridges over the River Turia: Genesis of the Urban History of Valencia"

_land, doi:10.3390/land12122175_

Round 1

Reviewer 1 Report

Comments and Suggestions for Authors

The paper is based on a multidisciplinary approach that combines urban history, hydrology, archaeology, and engineering. This is both its strength and its weakness. The paper explores the urban history of Valencia from the Roman to Muslim periods, focusing on the location of its bridges and the impact of their location on the urban and regional plan.  The cross-referencing of data from the various fields and the integration of their methodologies are of interest and value in this research. However, it is very difficult to ascertain the soundness of the data presented and the conclusions the authors draw from it.     

The article would benefit from reorganization, beginning with a clear outline of what is known from the archaeological literature on Valencia regarding its Roman bridges and roads, and where the lacunae exist. In the article’s current form, it is often difficult to separate existing literature from the authors’ speculation. Many sources are missing (what is known of the Roman city plan? What are the sources for the last section on the urban history of Valencia in the Roman period?) The article advances a hypothesis about the location of the bridge; this should be presented as such, rather than using a phrase such as “scenario resolution” that is described as “definitive.”

The figures need to be improved (see notes for specific comments). The English needs to be edited for clarity. Many sentences are unclear. There are several incomplete sentences. 

110-123  Expand into a narrative paragraph, not a list

164-166 Show these ancient cities on a map with the roads discussed

183  oxbow?

194 terminology: east/west instead of left/right branches

232-240 Not clear what has been established in the archeological literature vs what is a new hypothesis of the authors. Cite sources.

273  Sources of archeological findings?

277 “undoubtedly”- avoid such terms

296  “this would confirm” avoid such terms

341, 357 “everything points to the fact…” avoid such terms

472 (and elsewhere)  Cardo, not Cardus

575 no sources cited for this section.  Where is the data from?

489 not “solution”  not scenario resolution, not “definitive.”

.

529-537 expand into a paragraph, not point form

563 source for Casado

569 What is the evidence? This is speculative

Figures

Figure 1:  need a different map that also includes  the location of the Via Augusta and Via Herculea. The 19th-century conditions shown here are not relevant to the article.

Figure 4: remove yellow.

Figure 5a, Figure 10: What is the source of the city plan? Are the locations of the gates the authors' hypothesis? Not clear

Figure 6- label Via Augusta

Figure 16: source? Date?

Figure 7: photo obscures the aerial photo information at the crossing

-pull photo insets to the sides of the map

Figure 10: source? The photo of the model is too small; labels are illegible.

Figure 11:  photos obscure the aerial photo. Move photos to the sides.

Comments on the Quality of English Language

Editing is required for greater clarity.

Author Response

Thank you

Reviewer 2 Report

Comments and Suggestions for Authors

First, I would like to highlight the interest of the object of study of this article (rows 34-35), which can be understood as framed in the broader topic of the relationship between communication infrastructures and urban morphology, a matter of interest to Historical Urban Geography, Geographic Urban History, Urban Planning and Art History.

The development of the text is appropriate both from the point of view of the structure and presentation of the contents as well as formally.

In my opinion, some aspects could be improved to facilitate understanding for potential readers. Thus, replacing some literary descriptions integrated into the text, which are very difficult to follow without the help of a map, with a small cartographic representation, as happens, for example, in lines 221-225 and on many other similar occasions.

The cartography used is quite correct, although somewhat repetitive. Some of the planes represented could be integrated and unified and the rest deleted.

The presentation of results is correct, although in many cases they are speculations rather than proven or verifiable aspects. It is suggested that the discussion and conclusions obtained from the contrast between various hypotheses and scenarios be taken to a specific section or heading to clear the results section, the reading of which is somewhat cumbersome.

Regarding the conclusions, it is missing that the authors have not advanced a little more from a chronological perspective to identify and explain the changes in urban morphology in later periods, in the medium and long term, to explain the incidence of the bridges over the Turia in the emergence of suburbs and pieces of the extramural urban fabric as well as other forms of peri-urban urbanization, instead of limiting itself almost exclusively to the urban impact during the construction period of the infrastructures studied. It would not be difficult to complete the analysis carried out since there is a large amount of high-quality bibliography in the areas of urban history, historical urban planning and historical urban geography about the city of Valencia.

As for bibliographical references, many of them are repeated when the work of the same author is used in several citations. Do the authors consider it necessary to cite the same work several times in these cases? As an example, the 4 quotes from Otto's Teixidor study that occupy places 34, 35 and 36 in the references section, among many other examples. It is suggested to find an alternative by repeating the numbering in the text, with identification of the cited page, but not in the list of bibliographic references.

Author Response

Thank you

Round 2

Reviewer 2 Report

Comments and Suggestions for Authors

The authors have introduced changes that significantly improve the text proposed for publication.